



# Geochemistry of PM$_{10}$ over Europe during the EMEP intensive measurement periods in summer 2012 and winter 2013

Andrés Alastuey[1], Xavier Querol[1], Wenche Aas[2], Franco Lucarelli[3], Noemí Pérez[1], Teresa Moreno[1], Fabrizia Cavalli[4], Hans Areskoug[5], Violeta Balan[6], Maria Catrambone[7], Darius Ceburnis[8], José C. Cerro[9], Sébastien Conil[10], Lusine Gevorgyan[11], Christoph. Hueglin[12], Kornelia Imre[13], Jean-Luc Jaffrezo[14], Sarah R. Leeson[15], Nikolaos Mihalopoulos[16], Marta Mitosinkova[17], Jorge Pey[18], Jean-Philippe Putaud[6], Véronique Riffault[19], Anna Ripoll[1], Jean Sciare[20, 21], Karine Sellegri[22], Gerald Spindler[23], Karl Espen Yttri[2]

[1] Institute of Environmental Assessment and Water Research (IDAEA-CSIC), Barcelona, Spain
[2] NILU-Norwegian Institute for Air Research, Kjeller, Norway
[3] Dipartimento di Fisica e Astronomia and National Institute of Nuclear Physics (INFN), Sesto Fiorentino (Firenze), Italy
[4] European Commission – DG Joint Research Centre, Ispra, Italy
[5] Stockholm University, ACES, Stockholm, Sweden
[6] Hydrometeorologic State Service, Ministry of Ecology and Natural Resources, Chisinau, Moldova
[7] CNR, Institute of Atmospheric Pollution Research, Monterotondo Stazione (Rome), Italy
[8] School of Physics, National University of Ireland Galway, Galway, Ireland
[9] Laboratory of Environmental Analytical Chemistry, Illes Balears University, Spain
[10] ANDRA - DRD - Observation Surveillance, Observatoire Pérenne de l'Environnement, Bure, France
[11] Environmental Impact Monitoring Center, Yerevan, Armenia
[12] Empa, Swiss Federal Laboratories for Materials Science and Technology 8600 Dübendorf, Switzerland
[13] MTA-PE Air Chemistry Research Group, University of Veszprém, Veszprém, Hungary
[14] Laboratoire de Glaciologie et Géophysique de l'Environnement, UGA-CNRS, St Martin d'Hères Cedex, France
[15] Centre for Ecology and Hydrology (CEH), Bush Estate EH26 0QB, UK
[16] Environmental Chemical Processes Laboratory, University of Crete, Heraklion, Greece
[17] Slovak Hydrometeorological Institute, Bratislava, Slovak Republic
[18] Spanish Geological Survey. Zaragoza IGME Unit, Zaragoza, Spain
[19] Département Sciences de l'Atmosphère et Génie de l'Environnement (SAGE), Mines Douai, Douai, France
[20] Laboratoire des Sciences du Climat et de l'Environnment, Gif/Yvette, France
[21] The Cyprus Institute, Energy, Environment and Water Research Center, Nicosia, Cyprus
[22] Laboratoire de Météorologie Physique LaMP-CNRS/OPGC, Aubiere, France
[23] Leibniz Institute for Tropospheric Research (TROPOS) Leipzig, Germany

*Correspondence to*: Andrés Alastuey (andres.alastuey@idaea.csic.es)

**Abstract.** The third intensive measurement period (IMP) organised by the European Monitoring and Evaluation Programme (EMEP) under the UNECE CLTRAP took place in summer 2012 and winter 2013, with PM$_{10}$ filter samples concurrently collected at 20 (16 EMEP) regional background sites across Europe for subsequent analysis of their mineral dust content. All samples were analysed by the same or a comparable methodology. Higher PM$_{10}$ mineral dust loadings were observed at most sites in summer (0.5-10 µg m$^{-3}$) compared to winter (0.2-2 µg m$^{-3}$), with the most elevated concentrations in the southern- and easternmost countries, accounting for 20-40% of PM$_{10}$. Saharan dust outbreaks were responsible for the high summer





dust loadings at western and central European sites, whereas regional or local sources explained the elevated concentrations observed at eastern sites. The eastern Mediterranean sites experienced elevated levels due to African dust outbreaks during both summer and winter. The mineral dust composition varied more in winter than in summer, with a higher relative contribution of anthropogenic dust during the former period. A relatively high contribution of K from non-mineral and non-sea-salt sources, such as biomass burning, was evident in winter at some of the central and eastern European sites. The spatial distribution of some components and metals reveals the influence of specific anthropogenic sources on a regional scale: shipping emissions (V, Ni, and $SO_4^{2-}$) in the Mediterranean region, metallurgy (Cr, Ni, and Mn) in Central and Eastern Europe, coal combustion (As, Se, and $SO_4^{2-}$) in Eastern countries, and traffic (Cu) at sites affected by emissions from nearby cities.

## 1 Introduction

Mineral dust along with sea-salt aerosols are the major components of particulate matter (PM) mass in the atmosphere on a global scale (IPCC, 2007), playing a key role in the planet (Knippertz and Stuut, 2014). Mineral dust particles are generated mainly by wind erosion and soil resuspension in deserts and arid regions (e.g., Zhao et al., 2010; Kok, 2011), and their size distribution is characterised by a coarse size mode with a small fraction in the accumulation mode. Atmospheric residence time varies from less than 1 day to more than 1 week, depending on particle size and composition, but essentially on the effect of mesoscale and synoptic meteorology.

The magnitude of dust emissions to the atmosphere depends on the surface wind speed and soil-related factors such as texture, moisture and vegetation cover. Currently there is still a high uncertainty in the estimates of global dust emissions varying from 500 to 3000 Mt year-1 according to models (Huneeus et al., 2011). The major mineral dust sources from arid and semi-arid continental regions are located in subtropical areas in the Northern Hemisphere, and the Sahara-Sahel-Chad dust corridor is considered as the largest one of these (Prospero et al., 2002; Moreno et al., 2006; Engelstaedter et al., 2006). The proximity of Southern Europe to these large dust emitting areas and region-specific meteorological dynamics are responsible for the frequent transport of dust over Southern Europe and the subsequent deterioration of air quality (e.g., Bergametti et al., 1989; Querol et al., 1998 and 2009; Rodriguez et al., 2001; Kallos et al., 2006; Mitsakou et al., 2008; Pey et al., 2013).

Natural mineral dust mainly consists of silicate, carbonate, phosphate and oxide/hydroxide minerals derived from the erosion and weathering of rocks and soils. The chemical composition of airborne mineral dust depends on the geology of the source region, reflecting the composition of the parent soil of the source area. However, changes in chemical composition and physical characteristics of mineral dust may take place during transport due to the preferential deposition of coarse particles (Aluko and Noll, 2006; Scheuvens et al., 2013) and to the interaction with other particulate and gaseous pollutants (Rodriguez et al., 2011), resulting in coating or mixing of dust particles (Kandler et al., 2007; Levin and Ganor 1996). However, these interactions are not always large and the chemical modifications can be minor (Aymoz et al., 2004).



Mineral dust particles are also emitted by anthropogenic sources, such as agricultural activities, construction sites, mining, certain industrial activities such as the cement and ceramic industries, and road dust resuspension. However, there is a much variation in the contribution of such emissions depending on geographic location (Tegen et al., 2004). Whereas recent satellite observations suggest that the anthropogenic fraction of suspended mineral dust on the global scale represents only 20 to 25% of the total (Ginoux et al., 2012), on local or regional scales the anthropogenic contribution can rise to become the dominant source and so highly affect air quality (Querol et al., 2004).

Research on mineral dust is of great interest due to the impact of such particles on air quality, health and climate. Inhalation of mineral dust has potential adverse effects on health caused by particle composition, shape and particle size (Morman and Plumlee, 2014). Numerous scientific publications have demonstrated a relationship between the increase in PM10 concentration during Saharan dust events and severe health outcomes in the Southern European region (Pérez et al., 2008 and 2012; Diaz et al., 2012; Mallone et al., 2011; Stafoggia et al., 2013). For instance, Pérez et al. (2008 and 2012) evidenced that short-term exposure to PM during Saharan dust days is associated with both cardiovascular and respiratory mortality in Barcelona. However, as shown in the literature review by Karanasiou et al. (2012), the studies published show contradictory results regarding the health impact of Saharan dust outbreaks. Thus, further studies on chemical characterization, mixing state and potential toxicity of coarse particles transported from the Saharan desert are needed to clarify this issue.

Mineral dust can impact climate by different mechanisms, but mainly by interaction with radiative forcing and by modification of cloud properties (see references in Highwood and Ryder, 2014; Miller et al., 2014; and Nenes et al., 2014). Dust particles may scatter and/or absorb solar radiation and also act as ice nuclei. Net global radiative forcing of mineral dust is estimated as –0.1 (–0.3 to +0.1) W m–2 (IPCC 2013). The optical properties of dust particles depend on its mineralogical and chemical composition, with hematite having the largest absorption of UV and visible spectra (Lafon et al., 2006) among the inorganic components.

PM mass concentrations are routinely measured in national and regional Air Quality networks, such as the European environmental agency's (EEA) air quality programme (EC, 2008) and the European Monitoring and Evaluation Programme (EMEP) (UNECE, 2009; Tørseth et al., 2012). Some PM components such as SO42-, NO3-, NH4+, organic carbon (OC) and elemental carbon (EC) are also measured at sites included in the referred networks.

Even though the measurement of mineral dust is a compulsory part of the EMEP monitoring program (UNECE, 2009), and chemical speciation is recommended by the World Meteorological Organisation (WMO) Global Atmospheric Watch (GAW) (WMO 2003 and 2007), it is not generally included as a routine part of the monitoring programmes, probably due to the lack of a reference method and because in some areas it has been considered as a minor component of PM. In some studies, mineral components are estimated from the analysis of the soluble fraction of one or few mineral elements (Rodríguez et al., 2012). This may result in the underestimation (or even overestimation) of the mineral load, given that most minerals are water insoluble.



The chemical composition of mineral matter has been the topic of several studies conducted across Europe (Querol et al., 2001; Moreno et al., 2006; Nava et al., 2012; Lucarelli et al., 2010; Alemón et al., 2004). Putaud et al. (2010) compiled data on physical and chemical characteristics of PM10 and PM2.5, including mineral dust, from 60 sites across Europe, but different sampling techniques and analytical methodologies hamper the comparability. A review on methods for long-term in-situ characterisation of aerosol dust is presented in Rodríguez et al. (2012).

The EMEP task force of measurement and modelling (TFMM) periodically arranges Intensive Measurement Periods (IMP) as supplement to the continuous monitoring in EMEP (Aas et al., 2012). The third EMEP IMP took place during summer 2012 and winter 2013, and was organised in cooperation with the EU funded projects: Aerosols, Clouds, and Trace gases Research InfraStructure Network (ACTRIS), The Chemistry–Aerosol Mediterranean Experiment (ChArMEx) and Pan-European Gas-AeroSOls-climate interaction Study (PEGASOS). One of its major aims was to study the mineral dust and trace metal content of the PM10 fraction.

During the third EMEP IMP, PM10 filter samples were collected at 20 regional background sites (16 EMEP sites) representing different European rural background environments, by using an identical approach both with respect to sampling and subsequent speciation analysis of the mineral dust content. This qualifies for a unique data set which enables an extensive evaluation of sources, transport, and regional distribution of mineral dust across the European continent. In this paper spatial, temporal and chemical variations in mineral dust composition and trace metal concentrations are examined in order to reveal regional variations in background aerosol sources across a wide area of Europe.

## 2 Methods

### 2.1. Sampling sites

Ambient aerosol filter samples were collected at sixteen EMEP and four other regional background sites located in fourteen European countries, thus covering a wide range of the EMEP domain (Fig. 1 and Table 1). Most of these sites participated in both the summer (June 8th to July 12th 2012) and the winter (January 11th to February 8th 2013) IMP.

Northern Europe was represented by Aspvreten (SE12) in Sweden. The Atlantic and Northwestern European region included Mace Head (IE31) in Ireland, Auchencorth Moss (GB48) in Scotland and Harwell (GB36) in England, all with different marine/urban air mass influences. Puy de Dome (FR30), Revin (FR09), SIRTA (FR20), and OPE (FR22) in France represented central western Europe; Ispra in Italy (IT04), Payerne in Switzerland (CH02), Melpitz in Germany (DE44) represented the central European area. Eastern European stations included Amberd in Armenia (AM01), Leova (MD13) in Moldova, Starina (SK06) in the Slovak Republic, and K-Puszta in Hungary (HU02). In the south, the Spanish sites of Montsec (ES22), Montseny (ES1778), and Can Llonmpart, in Palma de Mallorca, (ESCLl) were taken as representative of Southwestern Europe, the Montelibretti (IT01) site in Italy represented Central Southern Europe, and Finokalia (GR02) in Crete (Greece) represented Southeastern Europe.





## 2.2 Sampling

Daily $PM_{10}$ sampling (24h) was performed from 8[th] June to 12[th] July 2012 during the summer IMP, and from 11[th] January to

8[th] February 2013 for the winter IMP. $PM_{10}$ samples were collected on Teflon filters (PALLFLEX, Pall Corporation Teflon Membrane Disc Filters cod. R2PJ047, 2 μm, 47 mm, 50/pkg) using low volume samplers (1-2.3 $m^3 h^{-1}$). One field blank was collected at each of the sampling sites for each IMP in additionally to the analysis of laboratory blanks.

At Montseny (ES1778), $PM_{10}$ filters were sampled concurrently by a DIGITEL high volume sampler (30 $m^3 h^{-1}$), loaded with Quartz microfiber filters (PALLFLEX, Pall Corporation QAO2500, 150 mm) to compare different methods for measuring mineral dust.

Two additional sites participated in the EMEP IMP, but $PM_{10}$ filters were sampled and analysed by different methods. At Montsec (ES22) $PM_{10}$ was collected on Quartz microfiber filters (PALLFLEX, Pall Corporation QAO2500, 150 mm) using a MCV/CAV-A MSb high volume sampler (30 $m^3 h^{-1}$), whereas at K-Puszta (HU02) $PM_{10}$ was collected on Quartz microfiber filters (PALLFLEX, Pall Corporation, 47 mm) by using low-volume samplers (1 $m^3 h^{-1}$).

Information regarding sampling and filter analysis is summarised in Table 1.

## 2.3. Chemical characterisation

Most samples collected during these campaigns were analysed by PIXE (Particle-Induced X-ray Emission). PIXE is a useful technique for the analysis of refractory elements in atmospheric aerosol samples. The convenience of PIXE for atmospheric aerosol research has been evaluated by Lucarelli et al. (2010) and Maenhaut (2015). This technique allows the simultaneous determination of concentrations for elements with atomic number higher than 10 with a good sensitivity. PIXE is a non-

destructive technique that does not require a specific sample preparation. The capability of analysing samples with very low concentrations without sample pre-treatment lowers the risk of contamination and analyte losses.

In the present study, samples andboth laboratory and field blank filters from all sites (with the exception of K-Puszta and Montsec) were analysed by PIXE with 3 MeV protons at the INFN LABEC (Laboratorio di Tecniche Nucleari per l'Ambiente e i Beni Culturali – Firenze, Italy) with an external beam set-up (extensively described by Lucarelli et al., 2014).

At LABEC, simultaneous high sensitivity detection of most mineral elements (Na, Mg, Al, Si, K, Ca, Ti, Mn, Fe, Sr, Zr) in a filter sample can be performed within a rather short time frame (30 sec to 3 min), depending on the aerosol load. The minimum detection limit for each element is listed in Table S1 in the Supplementary Information.

Samples collected by high-volume samplers at ES1778, ES22 and HU02 were analysed by ICP-AES and ICP-MS at the IDAEA CSIC laboratories, following the method devised by Querol et al. (2001). At ES1778, the correlation between the

concentrations obtained by PIXE and ICP-AES/MS was very high for most major and trace elements (Al, S, K, Fe, Ca, Ti and Mn), with $R^2 = 0.95$-0.99 and slopes close to 1 (1.01-1.05). Higher slopes were determined for Na (1.13) and Mg (1.26), although $R^2 > 0.9$. Correlations were lower for V, Cr, Pb and Sr ($R^2 = 0.60$-0.69; slopes 0.3 to 0.9), and very low for Ni ($R^2 =$





0.19, slope 0.4; see Fig. S1 and Table S2in the Supplementary Information), the concentrations of which were often close to the analytical detection limit.

The content of carbonate carbon was determined for 86 filter samples collected at 6 sites; i.e., ES1778, IT01, IT04, CH02, GR02 and MD13. The analysis followed the approach by Karanasiou et al. (2011), which by acidification (phosphoric acid) transforms the filter sample $CO_3^{2-}$ content into $CO_2$. $CO_2$ is subsequently determined by a flame ionization detector of a commercial thermal-optical analyser, after reduction to $CH_4$. The method detection limit was 0.2 µg C cm$^{-2}$.

### 2.4. Data treatment

Some elements such as Na, Mg, Ca and K are associated with both mineral dust and sea-salt aerosol. The sea-salt contribution was estimated for each element prior to estimating the mineral load. The concentrations of these elements in sea water are well known and therefore it is possible to estimate the marine contribution to their content in PM once we know the concentration of one of these elements. In marine-influenced and background areas, Cl$^-$ has a major marine origin, therefore the marine fraction of each element could be estimated from Cl$^-$ concentrations obtained for each sample. However,

this method may result in an underestimation of the marine aerosol due to the interaction (in the atmosphere or on the filter) of NaCl with acidic species resulting in the formation of HCl or $NH_4Cl$, which may be volatilised in the atmosphere (Harrison and Pio, 1983).

It is well known that Na aerosols commonly havea  marine origin, as halite associated with Cl$^-$, however they can also derive from a range of other minerals such as clays, carbonates, feldspars and sulphates. Thus, assuming an exclusively or even

dominantly marine origin for Na can cause significant errors, particularly in the Southern European countries that are frequently affected by dust events. Average Na/Cl ratios obtained for each site during the summer EMEP IMP varied from 0.5 for Mace Head (IE31), between 1-2 for GB48, IT01 and GR02, and from 3 to 9 for the remaining sites. Therefore, given that the Na/Cl sea water ratio is 0.56, there is a clear excess of Na (mineral or anthropogenic) and/or a depletion of Cl$^-$ (by volatilisation) for most sites, except for IE31.

The mineral fraction of Na was estimated from the content of Al by using the ratio determined by Moreno et al. (2006) for soils and dust in North Africa. Thus, the mineral sodium (non-sea-salt sodium, nssNa) was obtained by multiplying Al concentration by 0.12 (nssNa = [Al]x0.12). Hence, the sea-salt fraction of Na (ssNa$^+$) can be estimated subtracting nssNa from the total Na.

The sea-salt fraction of calcium, magnesium, potassium and sulphate was estimated by using their sea water ratios with

respect to ssNa$^+$ (Nozaki, 1997). Finally, the total sea-salt load was determined by the sum of Cl$^-$, ssNa$^+$, ssCa$^{2+}$, ssMg$^{2+}$, ssK$^+$ and ssSO$_4^{2-}$. The non-sea-salt (nss) fractions of Ca, Mg, Mn, Na, K and SO$_4^{2-}$ were obtained by subtracting the previously calculated sea-salt fraction from their bulk concentration.

Correlation between nssK and Al concentrations for southern sites during the summer Saharan dust events (SDE) permitted the identification of two major sources for nssK: a mineral (dustK) and a biomass burning (bbK) source. The mineral



fraction is estimated from Al (dustK = 0.31 * Al, Moreno et al., 2006), and the biomass fraction by the difference: bbK=nssK – dust K.

Finally, the total mineral dust concentration was determined by addition of the concentrations of all mineral related elements expressed as oxides, such as $Al_2O_3$, $SiO_2$, $Fe_2O_3$, $TiO_2$, $P_2O_5$, nssCaO, nssMgO, $nssNa_2O$, and $dustK_2O$.

Concentrations of elemental sulphur (S) were measured by PIXE. Assuming that most S is present as sulphate in PM,

concentrations of S, as measured by PIXE, were converted to $SO_4^{2-}$ multiplying by 2.995.

## 2.5. Saharan dust events (SDE)

The impact of African dust outbreaks on air quality over Europe during the periods of the study was traced by using publicly available information: i.e., dust maps supplied by the Navy Aerosol Analysis and Prediction System (NAAPS) model from the Navy research Laboratory (http://www.nrlmry.navy.mil/aerosol); data and/or images from the BSC-DREAM8b (Dust

REgional Atmospheric Model) model, operated by the Barcelona Supercomputing Center (BSC, http://www.bsc.es/projects/earthscience/BSC-DREAM/); and NASA Terra – MODIS satellite imagery (https://earthdata.nasa.gov/labs/worldview/).

During the summer 2012 IMP, two African dust outbreaks occurred: the first from the 17-23[rd] of June and the second one from the 28[th] of June to the 7[th] of July. The two episodes were initiated by the intense heating of the Sahara and the

consequent development of the North African thermal low south of the Atlas Mountains, coupled with anticyclonic conditions at upper atmospheric levels. This is the most frequent scenario causing dust outbreaks over Southwestern Europe (Moulin et al., 1998; Rodriguez et al., 2001; Escudero et al., 2007; Pey et al., 2013), with the convective system pumping dust up to 5000 m a.s.l. Once the dust is injected into the mid-troposphere it may be transported toward Western Europe by the eastern branch of the high (pressure) present over North Africa (Rodriguez et al., 2001). In such cases, the air masses are

heavily loaded with dust and are transported toward the north covering most of the western Mediterranean basin, forming a wide plume of dust. As shown in the satellite image (Fig. S2b), by June 25[th] 2012 Saharan dust had spread to the west across the Canary Islands and Madeira. The dust started blowing a couple of days earlier in Algeria and Mali, and travelled hundreds of kilometres toward the northwest. Over the Atlantic Ocean, the dust made a giant turn toward the east, in the direction of the Mediterranean Sea. As shown by the NAAPS model, the plume first hits the Iberian Peninsula, moving north

and eastwards successively affecting parts of Central, South Central, and Southeastern Europe (Fig. S2a).

Short but intense dust episodes occurred during the winter 2013 IMP, affecting mainly sites in the Eastern Mediterranean. This is the typical scenario in winter, when the development of low pressure systems gives rise to the rapid transport of dust at surface levels towards the east (Moulin et al., 1998), resulting in severe dust episodes in the Eastern Mediterranean region (Pey et al., 2013). As shown in Fig. S3, a dust plume blew off the coast of Libya and crossed the Mediterranean towards the

east on the 20[th] of January, reaching Greece and Eastern Europe by the 21[st] of January. Other episodes impacted the Eastern Mediterranean on the 3[rd] and the 7[th] of February. The satellite image corresponding to the 7[th] of February reflects the





occurrence of multiple dust plumes transported from the coast of Libya towards the northeast (Fig. S3). The transport at surface levels results in a low spatial dispersion of the plume.

## 3 Results

### 3.1 $PM_{10}$ levels

Concentrations of $PM_{10}$, measured by gravimetry or by automatic monitors (e.g. TEOM) were available for the samples analysed at most sites, with the exception of FR20, FR30 in winter and FR22 in summer. $PM_{10}$ concentrations in summer were, in general, higher in the southern sites (Fig. 2). Mean levels ranged from 20 µg m$^{-3}$ to 31 µg m$^{-3}$ at DE44, ES22, GR02 MD13, ES1778, and IT01; from 12 µg m$^{-3}$ to 20 µg m$^{-3}$ at CH02, FR09, AM01, IT04, HU02, SK06 and ESCLl; and from 3 µg m$^{-3}$ to 8 µg m$^{-3}$ at FR30, GB48 and SE12. The high levels measured at IT01 are probably due to its proximity to the city of Rome, whereas similarly high levels registered in ES1778 are attributed to the occurrence of two African dust outbreaks (see below).

In winter, low $PM_{10}$ levels were registered in Northern and Southwestern Europe (3-9 µg m$^{-3}$) and the highest $PM_{10}$ levels were measured in Southern, Eastern and Central Europe (20-40 µg m$^{-3}$). High $PM_{10}$ levels were recorded along a Northwest to Southeast transect, ranging from 22 µg m$^{-3}$ to 40 µg m$^{-3}$, with the highest concentrations observed for GR02, CH02 and IE31 (40, 37 and 36 µg m$^{-3}$, respectively).

### 3.2 Spatial distribution and elemental affinities

Average concentrations of major and trace elements determined for each site during the summer and winter IMPs are summarised in Tables S3 and S4. In the current section, the spatial variation of major elements is discussed. The elements were classified according to their main affinities into marine, mineral, or mixed origin groups.

### 3.2.1 Sea-salt-related elements

Sea salt (estimated as described in the methodological section, Sect. 2.4) accounts for a larger fraction of $PM_{10}$ at coastal sites, around 10% on average, with important variations in winter (Fig. 3). During the summer IMP, the average sea-salt load ranged from 1.9 µg m$^{-3}$ to 2.7 µg m$^{-3}$ at the coastal sites (10% of $PM_{10}$ at ESCLl and GR02 in the Mediterranean, 16% at IE31 in the Atlantic), from 0.2 µg m$^{-3}$ to 0.7 µg m$^{-3}$ at sites situated close to the coast (4-10% of $PM_{10}$ at IT01, GB48, ES1778 and SE12), and were below 0.2 µg m$^{-3}$ for the remaining sites. In winter, the sea-salt load was up to one order of magnitude higher, compared to the summer ones at the northwestern sites, reaching an average concentration of 19 µg m$^{-3}$ at the coastal IE31 (>50% of $PM_{10}$) and values greater than 2 µg m$^{-3}$ at GB48 and GB36 (30 and 14% of $PM_{10}$, respectively). In the Mediterranean area, the average sea-salt contribution was slightly higher at GR02 (3.9 µg m$^{-3}$, 10%$PM_{10}$) compared to the other sites (< 1 µg m$^{-3}$).



### 3.2.2 Mineral dust

In order to identify major sources of mineral dust in the study area, we have explored the correlations between elements with a major mineral affinity. We find that a number of mineral-related elements correlate strongly with Al and Si, whatever the site, suggesting a dominant aluminium-silicate (Al-Si) occurrence in PM. These components are Si, Ti, and Fe, and all showed higher concentrations at the southern and eastern sites (as deduced from data shown in Tables S3 and S4).

During the summer IMP, concentrations of Al were markedly higher in the Southwestern, South Central, and Eastern European regions, reaching average concentrations above 1 $\mu$g m$^{-3}$. These decreased northwards, being <0.1 $\mu$g m$^{-3}$ at the Northwestern and Northern European sites. High levels of Al recorded at the southern sites were related to the impact of Saharan dust events (as will be discussed in the following sections) and to resuspension of soil dust on a regional scale. In Eastern Europe, where Saharan dust events did not impact PM levels, a local/regional dust source is deduced. Concentrations of Al decreased at most sites in winter, being <0.2 $\mu$g m$^{-3}$ in the south and <0.1 $\mu$g m$^{-3}$ in the north. An exception was seen for GR02 (average of 2 $\mu$g m$^{-3}$) due to the impact of Saharan dust events even in winter. A similar spatial distribution was observed for Si, given its very high correlation with Al ($R^2$=0.98).

Figure 4 depicts cross-correlation plots between the $R^2$ coefficients and the slopes of the regression equation (y=ax+b) calculated for major elements at each site. The figure shows that there was a high correlation ($R^2$>0.8) between Si and Al (excluding IE31, $R^2$=0.23) when considering both IMPs, with $R^2$>0.95 at most sites. The slope of the equation (Si=xAl+b) was within 1.9 (IT04 and DE44) and 2.9 (MD13), ranging between 2.0 and 2.6 for most sites. Thus, the average Si/Al ratio was 2.34 (Si=2.34xAl). There was a spatial gradient, this being most evident in summer, which showed a downward trend of the Si/Al ratios from Eastern (slope=2.4-2.9) towards Western Europe (2.0-2.2). These high correlation coefficients and the Si/Al ratios point to a clay dominated source (mainly illite, Si/Al=2.5) for both elements. The relatively low Si/Al ratio seen for Southern and Southwestern Europe was probably due to a higher contribution of kaolinite (Si/Al=1.0). In winter, this spatial variability is not as clear, possibly due to the lower concentrations.

Aluminium also correlates well with Fe, with $R^2$>0.70 for most sites, pointing to a major aluminium-silicate association. The determination coefficient is $R^2$>0.90 for some sites when considering both IMPs (AM01, GR02, CH02, MD13, SK06, ES22, ES1778, IT01, FR30, FR22), confirming a mostly unique clay-related source for Fe (illite) at these sites. In these cases the slope varies from 0.5 (ES22, ES1778, FR30, FR22, SK06) to 0.7 (AM01, MD13) reflecting a different local mineral assemblage. Lower correlations were obtained for IE31 ($R^2$=0.01) and IT04 ($R^2$=0.15). At the other sites, $R^2$ ranged from 0.6-0.8, hence a significant contribution of other mineral or non-mineral sources can be suspected. The correlation was slightly higher in summer, with $R^2$ ranging from 0.7 to 1.0 at all sites, except at IE31 ($R^2$=0.10), reflecting the higher contribution of mineral dust. In winter, however, $R^2$ is lower than 0.8 for most sites, with slopes frequently >1, reflecting a lower contribution of mineral dust and probably a higher contribution of anthropogenic sources of Fe.

Calcium has a major mineral affinity, and is only a minor species in sea salt, being present in the atmosphere as carbonate (calcite $CaCO_3$, dolomite $CaMg(CO_3)_2$) or as calcium oxide (CaO), and to a minor extent as gypsum ($CaSO_4 \cdot 2H_2O$). This



element is usually related to natural sources (soil resuspension), although it can be emitted by a number of anthropogenic
sources, such as road dust and construction activities. Calcium carbonate may interact with acidic compounds in the
atmosphere forming coarse secondary calcium nitrates ($Ca(NO_3)_2$) and calcium sulphates ($CaSO_4 \cdot xH_2O$). The samples
analysed for carbonate Carbon (CC) had a rather consistent CC/Ca ratio considering all sites and regardless of the samples
being affected by African dust episodes or not (Fig. 5). When including all samples, the overall CC/Ca ratio was $0.12 \pm 0.01$
($R^2=0.75$) and the intercept $-0.03 \pm 0.01$ µCC m$^{-3}$. Assuming a method recovery of 100% (i.e. all CC present on the filter
sample is determined), the CC/Ca ratio value suggests that at least half of the Ca is present in PM$_{10}$ in other forms than
mineral carbonate, e.g. $Ca(NO_3)_2$ and $CaSO_4 \cdot xH_2O$. Given the high correlation between Ca and CC, it can be deduced that
non-carbonate Ca follows from the reaction between carbonates and acidic compounds.

### 3.2.3 Elements with a mixed source origin

Magnesium has a mixed origin and is associated with both marine aerosol and mineral dust. A dominantly marine origin was
deduced for Mg at the northern sites (IE31, SE12 and GB48), this element being highly correlated with Na ($R^2=0.81-0.98$)
and with a Na/Mg ratio ranging from 5.2 to 6.7, slightly lower than the ratio for sea salt (8.4, Drever, 1997). A mixed
marine/crustal origin was deduced for Southern and Central Europe with two different correlations between Na and Mg for
days with dust or marine influence. The nssMg may be associated with clays, carbonates (dolomite), aluminium-silicates or
salts. Considering all samples, the concentrations of nssMg were highly correlated with Al ($nssMgO=0.30 \times Al_2O_3$,
$R^2=0.76$). However, the correlation between nssMg and Al varied considerably between sites, reflecting different mineral
associations for nssMg. Three groups of sites can be distinguished as a function of $R^2$. The determination coefficient was
<0.3 at IE31, GB48 and SE12; >0.9 at AM01, CH02, GR02, MD13, SK06, ES1778 and FR30; and between 0.6 and 0.9 at
the remaining sites. For most sites, the slope ranged 0.15-0.2 (Fig. 4). At two eastern sites AM01 and GR02, a unique crustal
origin was deduced for nssMg, and with relatively high concentrations. In this region the Mg/Al ratio increased to 0.5
($R^2=0.98$), which could be related to the high Mg geochemical anomaly of the region. An intermediate slope (0.25) was
obtained for MD13.

Non-sea-salt potassium (nssK) has a major aluminium-silicate affinity and may be present in minerals such as feldspars and
clays, but can also be emitted during biomass combustion. When considering all samples, nssK and Al concentrations were
moderately correlated ($R^2= 0.47$), with a slope of 0.26 and a low intercept value (0.1 µg m$^{-3}$). In summer, the correlation was
higher ($R^2= 0.78$), indicating a major Al-Si affinity for nssK. Figure 4 shows this correlation varies considerably from site to
site, especially in winter. Most southern sites (ES1778, GR02, ESCLl, ES22, AM01, FR30) were characterised by high $R^2$
(0.8-1.0) and slopes between 0.2 and 0.3, confirming a major mineral origin. For Northern and Central European sites, as
well as for some southern sites (IT01), the determination coefficients were lower ($R^2<0.6$). In summer, this grouping is
obvious, but the slopes ranged 0.15-0.4 and $R^2=0.5-1$ (except for IE31) for all sites. In winter, there is a higher dispersion
(Fig. 4). High nssK/Al correlation ($R^2=0.8-1$) was only determined for ES1778, FR30 and GR02, with slopes 0.3-0.5. At the
remaining sites $R^2<0.6$ (ranging 0-0.6) and the slopes varied from 0.1-7. This indicates an additional potassium source in



winter, which most likely is biomass combustion. This additional source was more evident at certain sites (e.g., IT04, CH02, GB36, FR20, FR22, SE12, Fig. 4) when nssK/Al slopes >1 were recorded in winter, revealing a significant contribution from biomass burning. Thus, correlation of nssK with Al permitted us to identify two major sources for nssK, namely those of

dust (dustK) and biomass burning (bbK). For those sites with a high correlation ($R^2$>0.9) between nssK and Al, we consider that nssK is mostly associated with clays. In these cases we used the average ratio K/Al in summer to calculate dustK. This ratio varied from 0.2 at ES22, 0.3 at ES1778 and FR30, 0.4 at AM01 and FR22, and 0.5 at GR02, MD13 and SK06. For the remaining sites, we used an intermediate value (nssK/Al=0.3), which coincides with the average ratio obtained by Moreno et al. (2006) for Saharan dust samples and with the one obtained for samples collected during SDE in this study (see next

section). Once the dustK was estimated, the biomass fraction was calculated as the difference: bbK=nssK – dustK.

The spatial variation of the estimated dustK and bbK average concentrations in $PM_{10}$ is presented in Fig. 6. Concentrations of dustK were higher in summer, ranging from 10 ng m$^{-3}$ at the northern sites to 100-200 ng m$^{-3}$ at the southern sites and >400 ng m$^{-3}$ at MD13. In winter, dustK ranged 5-50 ng m$^{-3}$, except at GR02 with an average concentration of 400 ng m$^{-3}$, which is higher than in summer (200 ng m$^{-3}$) due to the occurrence of Saharan dust events in winter. By contrast,

concentrations of bbK were higher in winter, ranging from 50 to 500 ng m$^{-3}$, especially in Central and Eastern Europe. If we compare the relative contribution of bbK and dustK to the nssK, the dust contribution was most important in summer for the Mediterranean and the Eastern European sites, whereas for the Northern European sites dust and biomass were equally large sources. In winter, bbK clearly dominated, reflecting the impact of biomass combustion mainly at the northern and central sites. However, this estimation may be subject to significant errors for specific cases. Thus, the relatively high concentrations

of bbK estimated for IT01 and MD13 in summer could be due to a different local soil composition.

These findings demonstrate the influence of local (both natural and anthropogenic) and external emissions as revealed by the ratios between PM mineral components. These influences can result in significant errors when trying to estimate the PM mineral load by applying factors to the concentration of a single measured element, as proposed by a number of previously published works, an approach that should therefore be used with caution.

**3.3 Mineral dust contribution: impact of Saharan dust events**

Figure 7 shows the spatial variation of the average mineral dust load contribution determined for each site and IMP following the procedure explained in Sect. 2.4. In general, higher concentrations of mineral dust were recorded in summer compared to winter. The exceptions were GR02, strongly affected by Saharan dust events in winter, and IE31, with very low concentrations of mineral dust. The summer maxima of mineral dust were more evident in the southern and eastern

countries, showing a spatial distribution similar to that described for $PM_{10}$ (Fig. 2). The highest dust load (5 to 10 µg m$^{-3}$) was determined at the MD13, IT01, ES22, AM01 and ES1778 sites. Intermediate levels (between 2.5 and 5 µg m$^{-3}$) were obtained at GR02, ESCLl, HU02, SK06, FR09, FR22 and IT04, followed by CH02 and DE44 (1.5-2 µg m$^{-3}$). At the remaining sites (FR30, SE12, GB48 and IE31) the dust load was < 0.5 µg m$^{-3}$. Hence, the mineral load accounted for less



than 10% of $PM_{10}$ in summer at the northern sites (2% in IE31, 6% in SE12 and 9% in DE44), for 15% to 25% of $PM_{10}$ at most sites, and for more than 30% at ES22 (34%), MD13 (38%) and AM01 (42%).

The time series of the ambient air mineral dust concentration during the summer IMP is presented in Fig. 8. Mineral dust concentrations increased at southern and at certain central European sites during the two SDEs observed during the summertime IMP; i.e., from the $17^{th}$ to the $23^{rd}$ of June and from the $28^{th}$ of June to the $7^{th}$ of July. The mineral dust concentration first increased at the southwestern sites (ES22 and ES1778), reaching 12 µg m$^{-3}$ on the $17^{th}$ of June (Fig. 8). As the plume moved eastward (see Fig. S2), levels increased at the two Italian sites, reaching 15 µg m$^{-3}$ at IT01 on the $22^{nd}$ of June. On the $28^{th}$ of June a second and more intense plume was observed over the European continent, spreading to distant areas such as Germany and the British Isles. The second SDE first impacted the Iberian Peninsula, which experienced daily mineral dust concentrations of 34 µg m$^{-3}$ at ES22 and 20 µg m$^{-3}$ at ES1778 on the $28^{th}$ of June. Mineral dust levels decreased northwards, reaching maximum daily concentrations on the $28^{th}$ at CH02 (14 µg m$^{-3}$), ESCLl (11 µg m$^{-3}$), FR09, FR22 and DE44 (7-8 µg m$^{-3}$), FR30 and GB48 (3 µg m$^{-3}$). On the $29^{th}$, the concentrations of mineral dust increased at ES22 (up to 39 µg m$^{-3}$) and at ES1778 and started increasing at the Italian sites (IT01 and IT04). On this day, the concentration of mineral dust decreased at the mentioned central and north European sites (CH02, ESCLl, DE44, FR30 and GB48). On the $30^{th}$ of June, the mineral load started decreasing at ES22 but still increased at ES1778, reaching 35 µg m$^{-3}$. Subsequently, the mineral load decreased at these sites, but increased at IT04 and IT01, reaching 9 µg m$^{-3}$ on the $30^{th}$ of June, reflecting the transport of the dust plume towards the Eastern Mediterranean region. The first of July, levels of mineral dust reached the maximum at ES1778, IT01 and HU02 (12-14 µg m$^{-3}$). On the $5^{th}$ of July the mineral load peaked at GR02, reaching 15 µg m$^{-3}$. From the first of July and until the end of the June/July 2012 EMEP IMP, levels remained relatively high in the central and eastern parts of the Mediterranean Basin. During these dust episodes, the mineral dust concentration was always higher at ES22 than at ES1778, reflecting the transport of dust at high altitudes in summer. Finally, during the SDEs, the mineral dust contribution to $PM_{10}$ was on average 35% at the affected sites , ranging from 25% at DE44 to 55% at ES22.

It should be mentioned that at the Spanish sites the contribution of the Saharan mineral dust during account for two exceedances of the daily limit value of 50 µgPM$_{10}$ m$^{-3}$, established by the EC Directive (EC, 2008).

The SDEs only partly affected Eastern Europe. Thus, the mineral dust loading observed for this region is presumably related more to the influence of local and/or regional sources. The presence of distinctive geochemical ratios (e.g. Si/Al and Mg/Al) characteristic for this region is confirmed in the present study which has allowed us to demonstrate the importance of crustal sources in the Eastern European region. Further research is needed to identify these sources in more detail, e.g. to establish emission factors, and to quantify their impact on ambient PM levels over longer periods. Another complicating factor is that this area  is likely impacted at times by emissions of dust from the Arabian deserts.

In winter, the concentrations of mineral dust were lower for all sites except for GR02 (Fig. 7), where an average concentration of 13.5 µg m$^{-3}$ was calculated due to the influence of short but intense Saharan dust events (see Fig. S3). Levels between 1 and 2 µg m$^{-3}$ were measured at IT04, IT01, MD01, and FR20. At the remaining sites, the dust load was <1 µg m$^{-3}$. In contrast  to the summer IMP, the dust load did not reflect the temporal variation of $PM_{10}$ during the winter IMP.



For Central and Northern Europe, the mineral fraction accounted for less than 5% of $PM_{10}$ at most sites, whereas it ranged between 5 and 10% for the southern and eastern sites. The 34% contribution of mineral dust to $PM_{10}$ at GR02 was an exception. The lower mineral dust loads in winter were attributed to a lower impact of the Saharan dust outbreaks (affecting only the eastern part of the Mediterranean) and reduced soil resuspension in this period.

Figure 9 shows the time evolution of the mineral dust concentration calculated for $PM_{10}$ during the winter IMP. The dust load was low at most sites, but peaked frequently at GR02 due to the impact of short but severe dust episodes which severely deteriorate air quality in the Eastern Mediterranean area (Dayan et al., 2008; Querol et al., 2009; Pey et al 2013). Thus, daily concentrations of mineral dust > 50 µg m$^{-3}$ were recorded at GR02 during 3 days (ranging 60-83 µg m$^{-3}$). The lower spatial dispersion of most winter episodes is a consequence of the transport at surface levels. During 21$^{st}$ of January a dust plume blew off the coast of Libya and crossed the Mediterranean towards the east (see Fig. S3). This plume extended to areas in northern Italy and dust concentrations increased simultaneously at distant sites such as GR02, HU02, SK06 and DE44, although it is difficult to attribute this increase exclusively to the impact of the dust plume.

## 3.4 Local/regional mineral dust contribution: chemical composition

In the current section, days not impacted by Saharan dust were studied to characterise the chemical profiles of local dust sources and to quantify their contribution to PM. Average dust concentrations were calculated for the two measurement periods without impact of Saharan dust (NO-SDE, Fig. S4). The results show a clear spatial pattern, as the dust concentration decreases towards the north and the west. Hence, dust attributed to regional/local sources accounted for more than 25 % of $PM_{10}$ at the eastern sites MD13 (7 µg m$^{-3}$) and AM01 (6 µg m$^{-3}$; summer only). At the southern sites (Western, Central and Eastern Mediterranean) local/regional dust ranged between 1.5-4 µg m$^{-3}$, accounting for 8-15% of $PM_{10}$. The higher contribution at IT01 (4 µg m$^{-3}$, 15% of $PM_{10}$) is likely related to anthropogenic sources in the urbanized areas nearby the site. At most Central European sites, the mineral dust concentration ranged between 0.5 µg m$^{-3}$ and 2 µg m$^{-3}$, accounting for 5-10% of $PM_{10}$. Lower values were estimated for sites in Northern and Western Europe.

The chemical composition of mineral dust may vary considerably depending on the location, soil composition and influence of external sources (i.e., Saharan dust). As shown in Sect. 3.3, Saharan dust had a significant influence on $PM_{10}$ levels at the southern sites during the summer IMP. As dust can be mixed with other sources and transformed during transport, the current set of data provides a unique opportunity to investigate the impact of Saharan dust on the chemical composition of mineral dust and on the variation of dust composition during transport.

Chemical composition of the mineral dust, expressed as oxides, was averaged at each site for those days with a significant impact of Saharan dust (Fig. 10). During the winter IMP only GR02 was considered, as it was the only site clearly impacted for a significant period (more than 1 day), although other sites such as HU02, SK06, and MD13 were also briefly impacted. Figure 10 shows there was a steep gradient in the concentration of mineral dust dominated by African dust, with the highest values seen for the areas located close to the source region; i.e., the Eastern Mediterranean (in winter) and the Southwest of





Europe (summer). The relative composition (expressed as %, Fig. 10 bottom) is quite similar among all sites, with differences for GR02 in winter (GR02-W), IT01 and ESCLl, which have a larger contribution of CaO.

Ternary diagrams with average dust composition during Saharan episodes are presented in Fig. S5. With respect to $SiO_2$, $Al_2O_3$ and $Fe_2O_3$ (Fig. S5), the average composition was almost identical for all sites, indicating a similar silico-aluminous composition. However, for similar $SiO_2/Al_2O_3$ ratios, the contribution of CaO varies (Fig. S5), being higher during the

winter (GR02-W) than during the summer events. This is probably related to the different source areas. In summer (see Fig.s S2 and S3), dust was emitted from South Algeria and Mali, whereas in winter dust blew off the coast of Libya. This is in agreement with previous studies showing higher content of Ca-carbonates for dust coming from Northeastern Africa (Formenti et al., 2011; Scheuvens et al., 2013). Relatively higher contents of CaO were also measured at ESCLl and IT01. This could be related to local contribution from soil resuspension given that the geology in these areas is characterised by the

presence of carbonate rocks (limestone). At IT01, an important fraction of calcium could be also related to the contribution of local anthropogenic sources, given its proximity to Rome.

Southern sites (orange and red colours) tend to have a higher content of CaO and MgO and a relatively lower content of $K_2O$ and $Fe_2O_3$ compared to Central and Northern European sites. This could be attributed to the different composition of local dust or to the preferential settling during transport due to the particle size and morphology of specific minerals. Again,

higher contents of CaO were obtained for IT01, GR02-W and ESCLl but whereas ratios of MgO/CaO keep constant for GR02-W and ESCLl, indicating a soil-related source (carbonates), CaO/MgO was higher at IT01, probably due to an anthropogenic contribution (construction, demolition, road dust).

Ternary diagrams based on the average composition of samples not affected by Saharan dust are shown in Fig. S6. As for the African dust episodes, the $SiO_2/Al_2O_3$ ratio was fairly constant for all sites except IE31. The possibility of the low levels

observed at IE31 affecting the results should not be excluded. For the other components, there is a wider variation in comparison with the African episodes (see Fig. S5). When plotting $SiO_2$, $Al_2O_3$ and CaO, the contribution of CaO clearly decreases from the southern to the eastern and northern sites. The obvious presence of this trend in the high altitude sites (ES22, AM01 and FR30) emphasises that it is a likely marker for far-travelled particles derived from the local/regional geology in the source areas. An inverse pattern is observed when plotting $SiO_2$ and $Al_2O_3$ with either $Fe_2O_3$ or $K_2O$ ; there is

a clear increasing trend of the relative contribution of $K_2O$ and $Fe_2O_3$ from the southern to the eastern and northern sites. However, in this case the average concentrations are very similar at the three high altitude sites, suggesting that this trend indicates the different influence of other sources such as biomass burning for $K_2O$ or iron steel industry for $Fe_2O_3$.

## 3.5 Sulphate

Sulphur is typically present in ambient air as sulphate ($SO_4^{2-}$). Therefore, although PIXE measures concentrations for S, we

have chosen to use sulphate in the present text. Non-sea-salt sulphate is a major secondary component formed by the oxidation of $SO_2$ emitted by combustion processes. Moreover, non-sea-salt sulphate ($nssSO_4^{2-}$) may have a minor mineral association, mainly as coarse gypsum, and can also be released from marine biogenic and volcanic emissions.



Concentrations of nssSO$_4^{2-}$ differed both spatially and temporally (Fig. 11). The highest mean SO$_4^{2-}$ concentration in summer was observed at the southeastern site GR02 (5.8 μg m$^{-3}$). Relatively high concentrations were also determined for the eastern, south central, and southwestern sites, ranging between 2.6-3.4 μg m$^{-3}$. Lower levels were measured at central and northern sites (0.5-1.1 μg m$^{-3}$) with an intermediate concentration for DE44 (2.0 μg m$^{-3}$). In winter, higher concentrations of SO$_4^{2-}$ were observed in Eastern and Central European sites (ranging 2-3.8 μg m$^{-3}$), whereas the average concentration was <2 μg m$^{-3}$ for Southern and Northern Europe.

Given the significant impact of Saharan dust outbreaks in southern European countries, it can be speculated that a minor fraction of SO$_4^{2-}$ could be primary gypsum, despite that it is a minor (usually <2%) component of Saharan dust, with particularly low concentrations in most North African source areas (Claquin et al., 1999; Scheuvens et al., 2013; Journet et al., 2014). To identify the contribution of primary gypsum to the observed concentrations of SO$_4^{2-}$, we have investigated the correlation of SO$_4^{2-}$ with major mineral elements. Figure 12 presents ternary diagrams for average concentrations of SiO$_2$, Al$_2$O$_3$ and SO$_4^{2-}$ for SDE and NO-SDE samples. For SDE the spatial pattern was different from that observed for Ca (Fig. S5). The highest relative SO$_4^{2-}$ contributions were obtained for GR02-S (summer) and central European sites, with lower contributions for GR02-W (winter) and some southern European sites, whereas higher relative contribution of Ca were determined for GR02-S and southern sites. A similar but opposite spatial trend of Ca was observed during NO-SDE (Fig. 12, right, and Fig. S6). Figure S7 presents the correlation between Al$_2$O$_3$ and SO$_4^{2-}$ for Saharan-dust-influenced samples at different sites. No correlation between these two species was observed for the mountain sites (FR30 and ES22), whereas the correlation was higher at distant sites, where a relative increase of the SO$_4^{2-}$ concentration was observed. The low concentrations of SO$_4^{2-}$ at the mountain sites, poorly correlated with Al$_2$O$_3$, indicate a low contribution of primary gypsum. The increase in concentrations of SO$_4^{2-}$ at distant sites from North Africa even during SDE suggests a major anthropogenic source, most probably related to the impact of regional SO$_2$ emissions, and originating mainly from fuel oil and coal combustion. A major source of SO$_4^{2-}$ in the Mediterranean region can be related to shipping (heavy oil) emissions, whereas in Central and Eastern Europe high concentrations are more often related to stationary sources, such as individual coal heated ovens and coal power plants, discussed in more detail in the next section.

### 3.6 Trace elements

Figures 13 and 14 show the spatial variation of selected trace elements during the summer and winter IMPs. Concentrations varied substantially between sites, whereas there was typically a minor difference with respect to concentration between summer and winter at most sites.

### 3.6.1 Mineral related elements

Some metals, such as Ti (shown in Fig. 13), Sr and Rb, were highly correlated with major mineral elements owing to their dominantly crustal origin. Consequently, these elements showed a spatial distribution similar to that of mineral dust, with higher mean concentrations in the southern and eastern European sites during summer, and at GR02 during winter. Mean




concentrations of these crustal-related elements were typically substantially lower in winter compared to summer. High concentrations of Sr at IE31 in winter were related to its partial marine origin, whereas its presence at IT04 was probably related to resuspension of road dust. The highest Sr concentrations in summer were observed at IT01, reflecting the influence of the anthropogenic emissions from the nearby city of Rome. In general, having a major carbonate affinity, Sr concentrations reflect the spatial distribution of Ca.

**3.6.2 Combustion related elements**

Vanadium (V) is a typical tracer of heavy oil combustion, usually associated with Nickel (Ni) and $SO_4^{2-}$ (Viana et al., 2008; Alleman et al., 2010). Elevated concentrations (5-6 ng m$^{-3}$) of V were recorded at the southern sites in summer, whereas V was <2 ng m$^{-3}$ at the remaining sites. Concentrations of V decreased in winter, being relatively higher at coastal sites such as IE31 (1.9 ng m$^{-3}$) and GR02 (3.4 ng m$^{-3}$). For the remaining sites, the mean V concentration was between 0.5 and 1 ng m$^{-3}$.

Nickel is commonly associated with V and is used to trace combustion of fuel oil, however it is also emitted by metallurgical processes (iron and steel manufacturing) (Viana et al., 2008; Pandolfi et al., 2011). In summer, the highest levels of Ni were recorded at southern (2-2.5 ng m$^{-3}$) and eastern European sites (1-2 ng m$^{-3}$, with a maximum value of 3.3 ng m$^{-3}$ at HU02), being < 1 ng m$^{-3}$ at the other sites. In winter, average concentrations were lower, with maximum values at the Italian sites (1.2-1.9 ng m$^{-3}$). At the other sites, the concentration ranged between 0.4 ng m$^{-3}$ and 1.1 ng m$^{-3}$, whereas it was < 0.4 ng m$^{-3}$

at the mountain sites.

Figure S8 shows the correlation between the average concentrations of V, Ni, and $SO_4^{2-}$, and the V/Ni ratio obtained for the two IMPs, and the correlation coefficients and the slopes of the regression equation calculated for these elements at each site. At the Mediterranean sites (red and orange colour), which experienced the highest concentrations of V, correlations were significant ($R^2$>0.4, p<0.001) when considering V and $SO_4^{2-}$, as was V versus Ni ($R^2$>0.4, p<0.0001). The V/Ni ratio ranged

from 2.3 to 2.5 (except for IT01; 1.5), which is within the range (2-4) identified for shipping emissions reported by Viana et al. (2008), Alleman et al. (2010) and Pandolfi et al. (2011). At AM01, V was found to correlate with Ni and $SO_4^{2-}$, however, the V/Ni and V/$SO_4^{2-}$ ratios were totally different from those obtained at the Mediterranean sites pointing to a different fuel combustion source.

Hence, a major common source of $SO_4^{2-}$, V and Ni, likely related to fuel oil combustion emissions in the Mediterranean has

been identified. As shown in Fig.s 11 and 13, there is a clear seasonal trend for these elements at the Mediterranean sites, with higher concentrations in summer. This may be attributed to lower maritime traffic in winter (cruises) and the higher photochemical oxidation of $SO_2$ in summer. Some Central and Eastern European sites, showed relatively high $SO_4^{2-}$ concentrations, but low concentrations of V, as well as a low correlation ($R^2$<0.4) between the two, indicating another major source for $SO_4^{2-}$.As shown in Fig. 14, summer average concentrations of As, a typical tracer for coal emissions (Pacyna,

1986) were higher at southeastern and central European sites (0.6-0.7 ng m$^{-3}$) with the most elevated levels at AM01 (1.6 ng m$^{-3}$). Lower values were recorded for western European sites, for which average concentrations ranged from 0.1 to 0.3 ng m$^{-3}$. In winter (Fig. 13), the higher concentrations were obtained at central and eastern European sites (0.6-1.3 ng m$^{-3}$). When



considering the two campaigns (Fig. S9), higher correlation coefficients between As and $SO_4^{2-}$ ($R^2$=0.42, p<0.0001) were determined for DE44, with relatively high concentrations of As, and for ES22, with very low As concentrations. At the

remaining sites, $R^2$ coefficients range 0.1-0.3. For similar $R^2$, higher As concentrations were recorded at the central and eastern sites (Fig. S9). It can be concluded that there is a higher contribution of coal combustion sources in Central and Eastern Europe (mainly in winter), that the high levels of As registered at AM01 could be related to other sources, and that the sporadic impact of coal power plant emissions at ES22 cannot be discarded.

### 3.6.3 Other industrial sources and road traffic

Tracers of industrial activities, such as Cr, Mn and Ni, do not show a clear spatial distribution pattern (see Fig.s 13 and 14). These elements may also have a mineral association and, as discussed for Ni, can be emitted by different processes. The higher concentrations of Cr, usually considered as a tracer for metallurgical activities (Querol et al., 2007), were measured in SK06 (4.4 ng m$^{-3}$) in summer and at IT01 (3.7 ng m$^{-3}$) in winter. At SK06 and AM01, concentrations show a very high correlation with Ni ($R^2$=0.94) confirming a common origin related to metallurgical activities. Relatively high determination

coefficients between Ni and Cr were also obtained at MD13, IT04, CH02, IE31, and ES22. At the MD13 and AM01 sites, high correlations of Cr and Ni with Mn and Cu, reinforce the link with metallurgical activities.

Although emitted in large proportion by industrial sources, Cu is often a tracer of non-exhaust vehicle emissions (Schauer et al., 2006; Amato et al., 2009). In summer this element was found to be present in higher concentrations at the southern and central European sites with average concentrations of 11 ng m$^{-3}$ at IT01 and 8.8 ng m$^{-3}$ at IT04 (Fig. 14). At the other sites,

mean Cu concentrations mostly range from 1 to 4 ng m$^{-3}$. In winter, high concentrations were again measured at IT04 and IT01 (15.9 and 5.7 ng m$^{-3}$, respectively) and at FR20 (9.4 ng m$^{-3}$). The lowest concentrations were measured at the mountain sites (ES22 and FR30, <0.5 ng m$^{-3}$).

To illustrate the differences in metalliferous tracers of emissions from major sources such as road traffic, fossil fuel combustion and industry, two ternary diagrams are proposed to compare the sites (Fig. S10). In these diagrams Cu is

considered to be a tracer of traffic, V (and partially Ni) as tracers of shipping emissions, As of coal combustion emissions and Cr and Ni as industrial (metallurgy) tracers. The relative proportions of these four possible sources allow us to differentiate (1) Mediterranean sites together with IE31, with a higher influence of fuel combustion (shipping) emissions; (2) sites with a relatively higher influence of traffic (IT04, IT01 and FR20) located close to important cities or busy roads; (3) the eastern (AM01, SK06, HU02, MD13) and central (DE44) sites with a higher impact of coal combustion emissions; (4)

some central and eastern sites more influenced by metallurgical activities. The mountain sites form a separate sub-group, having low concentrations of all of these tracers.



## 4 Conclusions

The third EMEP intensive monitoring period, conducted in summer 2012 and winter 2013, addressed the chemical speciation in $PM_{10}$ with a particular emphasis on mineral dust and trace metals. For the first time, mineral dust was determined in filter samples ($PM_{10}$) collected concurrently at a substantial number (20) of regional background sites across Europe, using a similar methodology at 18 of the sites; i.e. Particle-Induced X-ray Emission (PIXE), conducted at the INFN LABEC of Firenze, Italy. PIXE analysis allowed for the simultaneous detection of most mineral elements (i.e., Na, Mg, Al, Si, K, Ca, Ti, Mn, Fe, Sr, Zr) with high sensitivity.

The $PM_{10}$ mineral dust composition across Europe demonstrated the influence of both local (natural and anthropogenic) and external sources, as evidenced by ratios of different mineral components in the PM filter samples. In general, higher concentrations of mineral dust were recorded in summer compared to winter. The summer maxima of mineral dust were more evident in the southern and eastern European countries. The highest average dust load (5 to 10 $\mu g\ m^{-3}$) was determined at eastern and southwestern sites (MD13, IT01, ES22, AM01 and ES1778) accounting for 20-40% of $PM_{10}$. Intermediate levels (between 2.5 and 5 $\mu g\ m^{-3}$) were obtained at GR02, ESCLl, HU02, SK06, FR09, FR22 and IT04, followed by CH02 and DE44 (1.5-2 $\mu g\ m^{-3}$, 14-25% of $PM_{10}$). At the remaining sites (FR30, SE12, GB48 and IE31), the dust load was <0.5 $\mu g\ m^{-3}$ (2-10% of $PM_{10}$). In winter, the concentrations of mineral dust were lower (<1-2 $\mu g\ m^{-3}$, <5-10% of $PM_{10}$) for all sites except for GR02, where an average concentration of 13.5 $\mu g\ m^{-3}$ (34% of $PM_{10}$) was calculated due to the influence of short-lived, but intense, Saharan dust events.

Mineral dust was attributed to different origins. Saharan dust outbreaks were responsible for increases in mineral dust levels at sites in Southern and Central Europe in summer, whereas high levels of mineral dust at eastern European sites were attributed to local or regional sources. More specifically, the influence of two African dust outbreaks on the levels and composition of $PM_{10}$ across Europe was clearly detected. Contribution of mineral dust during SDEs may account of exceedances of the daily limit value established by the European Directive on air quality (EC, 2008) at MSY and MSC, in summer, and at GR02 in winter. The comparison between the average mineral loads estimated for the whole period at each site and for the non-SDE permitted us to estimate the Saharan dust contribution, this ranging across 0.1-0.4 $\mu g\ m^{-3}$ at the northern sites affected by the SDE, to 0.5-5 $\mu g\ m^{-3}$ at the southern sites. The impact of these Saharan dust intrusions resulted in a relative increase of $SiO_2$ and $Al_2O_3$ and a relative decrease of CaO, $K_2O$ and MgO. The composition of African dust affecting different regions of Europe was quite homogeneous over the different areas, showing that changes during dust transport were limited for this study.

The inherent complexity of different dust sources and composition can result in significant errors if the PM mineral load is estimated by applying factors to the concentration of a single measured element, as proposed by a number of studies. In our case, a higher variability of the dust composition was evident in winter, this being partially attributed to the higher relative contribution of anthropogenic dust at this time of the year. The relative contribution of K was also more pronounced in winter at some central and eastern European sites, probably reflecting a higher impact from biomass combustion.



Finally, the spatial distribution of trace metals and sulphate enabled the identification of specific anthropogenic sources at a regional scale: i.e., shipping emissions in the Mediterranean region (V, Ni, and $SO_4^{2-}$), metallurgy (Cr, Ni, and Mn) in Central and Eastern Europe, coal combustion sources (As, Se, and $SO_4^{2-}$) in eastern European countries, and traffic (Cu) at sites affected by emissions from nearby cities.

**Acknowledgements**

The present work was supported by the Co-operative Programme for Monitoring and Evaluation of the Long-range Transmission of Air pollutants in Europe (EMEP) under UNECE. We also acknowledge support by the European Union Seventh Framework Programme (FP7/2007-2013) through ACTRIS (grant agreement no. 262254).

The participation of IDAEA-CSIC was supported by the Spanish Ministry of Economy and Competitiveness and FEDER funds under the project PRISMA (CGL2012-39623-C02-1), by the Generalitat de Catalunya (AGAUR 2015 SGR33 and the

DGQA).

The authors wish to thank the Norwegian Ministry of foreign affairs for support to the measurements in Armenia and Moldova. The French participation to the campaigns was funded by the French Agency of Environment and Energy Management (ADEME, grants 1262C0022 and 1262C0039). Mines Douai acknowledges support from the CaPPA project which is financed by the French National Research Agency (ANR) through the PIA (Programme d'Investissement d'Avenir)

under contract ANR-11-LABX-0005-01, the "Nord-Pas de Calais" Regional Council and the European Regional Development Fund (ERDF). Participation of the ANDRA is also acknowledged for the measurements at OPE-ANDRA. The authors are also indebted to the fieldwork teams at Auchencorth Moss and Harwell (NERC CEH and Riccardo Energy and Environment Staff).

The authors would also like to express our gratitude to the Atmospheric Modelling Laboratory from the Barcelona

Supercomputing Centre, the Naval Research Laboratory and the SeaWiFS project (NASA) for the provision of the DREAM, NAAPs aerosol maps and the satellite imagery, respectively.

The authors wish to thank D. C. Carslaw and K. Ropkins for providing the Openair software used in this paper (Carslaw and Ropkins, 2012; Carslaw, 2012).

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





**Table 1. Sampling sites participating in the 2012-2013 IMP. S: summer IMP; W: winter IMP.**

| | Site | Name | Country | Altitude m a.s.l | Coordinates | Number samples | IMP | Sampler | Flow | Filter | Analysis |
|---|---|---|---|---|---|---|---|---|---|---|---|
| 1 | AM01 | Amberd | Armenia | 2080 | 40°23'04"N 44°15'38"E | 32 | S | Leckel | 2.3 m³ h⁻¹ | 47 mm P | PIXE |
| 2 | CH02 | Payerne | Switzerland | 489 | 46°48'47"N 05°56'41"E | 40 29 | S W | Leckel | 2.3 m³ h⁻¹ | 47 mm P | PIXE |
| 3 | DE44 | Melpitz | Germany | 86 | 51°31'48"N 12°55'48"E | 37 28 | S W | Partisol | 1 m³ h⁻¹ | 47 mm P | PIXE |
| 4 | ES1778 | Montseny | Spain | 740 | 41°46'00"N 02°21'00"E | 41 31 | S W | Partisol/DIGITEL | 1/30 m³ h⁻¹ | 47/150 mm P | PIXE/ICPs |
| 5 | ESCLl | Can Llompart | Spain | 45 | 39°50'16"N 03 01'25"E | 17 | S | IND LSV | 2.3 m³ h⁻¹ | 150 mm P | PIXE |
| 6 | ES22 | Montsec | Spain | 1570 | 42°03'05"N 00°43'46"E | 28 22 | S W | MCV-A/MSb | 30 m³ h⁻¹ | 150 mm P | ICPs |
| 7 | FR09 | Revin | France | 390 | 49°54'28"N 04°37'48"E | 7 6 | S W | Partisol | 1 m³ h⁻¹ | 47 mm P | PIXE |
| 8 | FR30 | Puy de Dôme | France | 1465 | 45°46'00"N 02°57'00" E | 22 12 | S W | Partisol | 1 m³ h⁻¹ | 47 mm P | PIXE |
| 9 | FR20 | SIRTA | France | 162 | 48°42'36"N 02°08'53"E | 8 | W | Partisol | 1 m³ h⁻¹ | 47 mm P | PIXE |
| 10 | FR22 | OPE | France | 392 | 48°33'44"N 05°30'21" E | 7 9 | S W | Partisol | 1 m³ h⁻¹ | 47 mm P | PIXE |
| 11 | GB36 | Harwell | England, UK | 137 | 51°34'23"N 01°19'00"W | 26 | W | Partisol | 1 m³ h⁻¹ | 47 mm P | PIXE |
| 12 | GB48 | Auchencorth Moss | Scotland, UK | 260 | 55°47'36"N 03°14'41"W | 32 20 | S W | Partisol | 1 m³ h⁻¹ | 47 mm P | PIXE |
| 13 | GR02 | Finokalia | Greece | 250 | 35°19'00"N 25°40'00"E | 39 30 | S W | Leckel | 2.3 m³ h⁻¹ | 47 mm P | PIXE |
| 14 | HU02 | K-Puszta | Hungary | 125 | 46°58'03"N 19°33'11"E | 39 26 | S W | Partisol | 1 m³ h⁻¹ | 47 mm P | ICPs |
| 15 | IT01 | Montelibretti | Italy | 48 | 42°06'00"N 12°38'00"E | 39 29 | S W | Swam 5a dual Chanel/FAI Inst. | 2.3 m³ h⁻¹ | 47 mm P | PIXE |
| 16 | IT04 | Ispra | Italy | 209 | 45°48'00"N 08°38'00"E | 37 29 | S W | Leckel | 2.3 m³ h⁻¹ | 47 mm P | PIXE |
| 17 | IE31 | Mace Head | Ireland | 5 | 53°19'36"N 09°54'14"W | 31 14 | S W | Partisol | 1 m³ h⁻¹ | 47 mm P | PIXE |
| 18 | MD13 | Leova II | Moldavia | 156 | 46°30'00"N 28°16'00"E | 44 29 | S W | Leckel | 2.3 m³ h⁻¹ | 47 mm P | PIXE |
| 19 | SE12 | Aspvreten | Sweden | 20 | 58°48'00"N 17°23'00"E | 26 29 | S W | Leckel | 2.3 m³ h⁻¹ | 47 mm P | PIXE |
| 20 | SK06 | Starina | Slovak Rep. | 345 | 49°03'00"N 22°16'00"E | 33/29 29 | S W | Partisol | 1 m³ h⁻¹ | 47 mm P | PIXE |





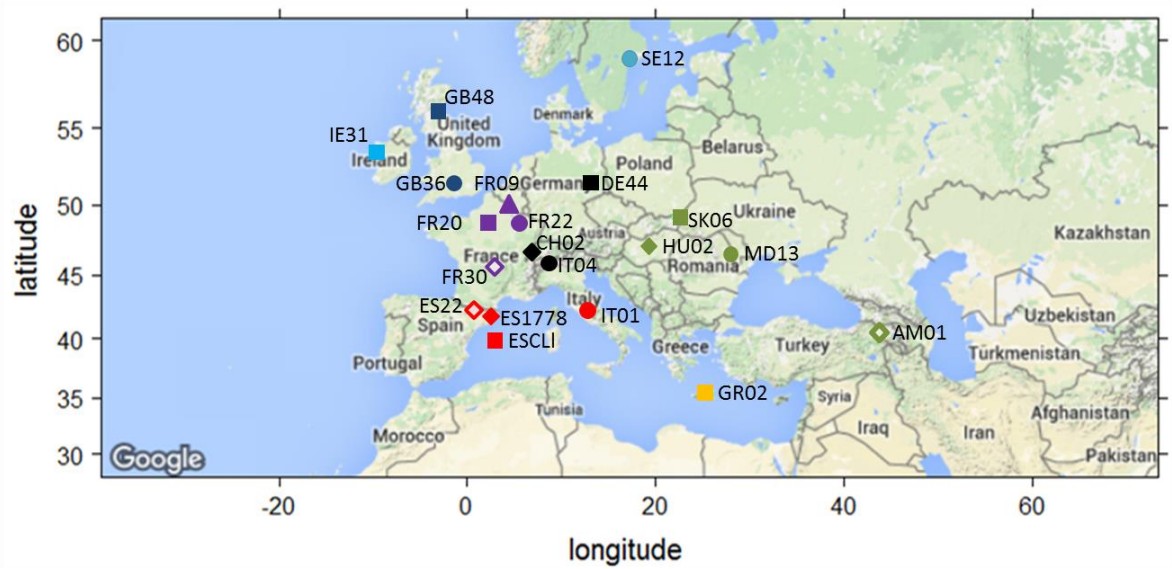

**Figure 1: Location of the sampling sites participating in the EMEP IMPs in summer 2012 and/or winter 2013. Blue symbols: Northern Europe; purple symbols: Central Western Europe; black symbols: Central Europe; green symbols: Eastern Europe; red symbols: Southwestern Europe; yellow symbols: Southeastern Europe; empty symbols: mountain sites.**

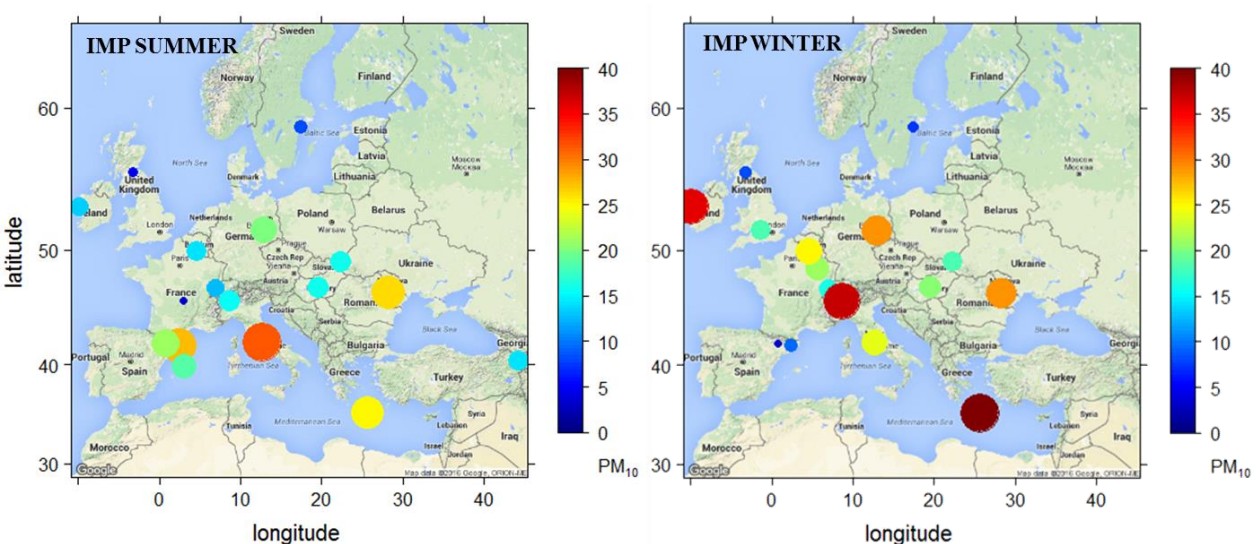

**Figure 2: Mean PM$_{10}$ concentrations (µg m$^{-3}$) recorded during the June/July 2012 and January/ February 2013 EMEP IMPs. The diameter of the circles is proportional to the concentrations.**




Figure 3: Spatial distribution of the mean sea-salt aerosol concentration (µg m⁻³) and its relative contribution (%) to PM₁₀ during the EMEP IMP in summer 2012 and winter 2013. The diameter of the circles is proportional to the concentrations.



**Figure 4: Scatter plots of determination coefficient vs slope for concentrations of selected elements at the sampling sites.**





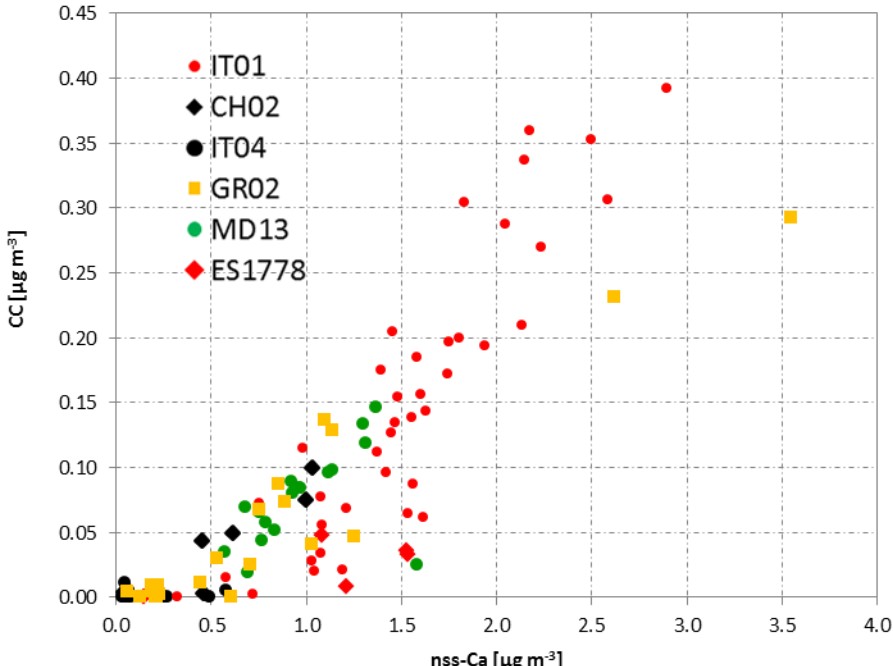

**Figure 5: Cross correlation plot between carbonate carbon (CC) and nssCa content (in μg m⁻³) in PM₁₀ at sites in different regions.**



Figure 6: Spatial distribution of the estimated average concentrations (in ng m⁻³) of the mineral (dustK) and the biomass burning (bbK) potassium at each site in the summer 2012 and winter 2013 IMPs. The diameter of the circles is proportional to the concentrations.







**Figure 7: Spatial distribution of the average mineral dust concentration (µg m⁻³) and its relative contribution (%) to PM₁₀ obtained at each site in the summer 2012 and the winter 2013 IMPs. The diameter of the circles is proportional to the concentrations.**



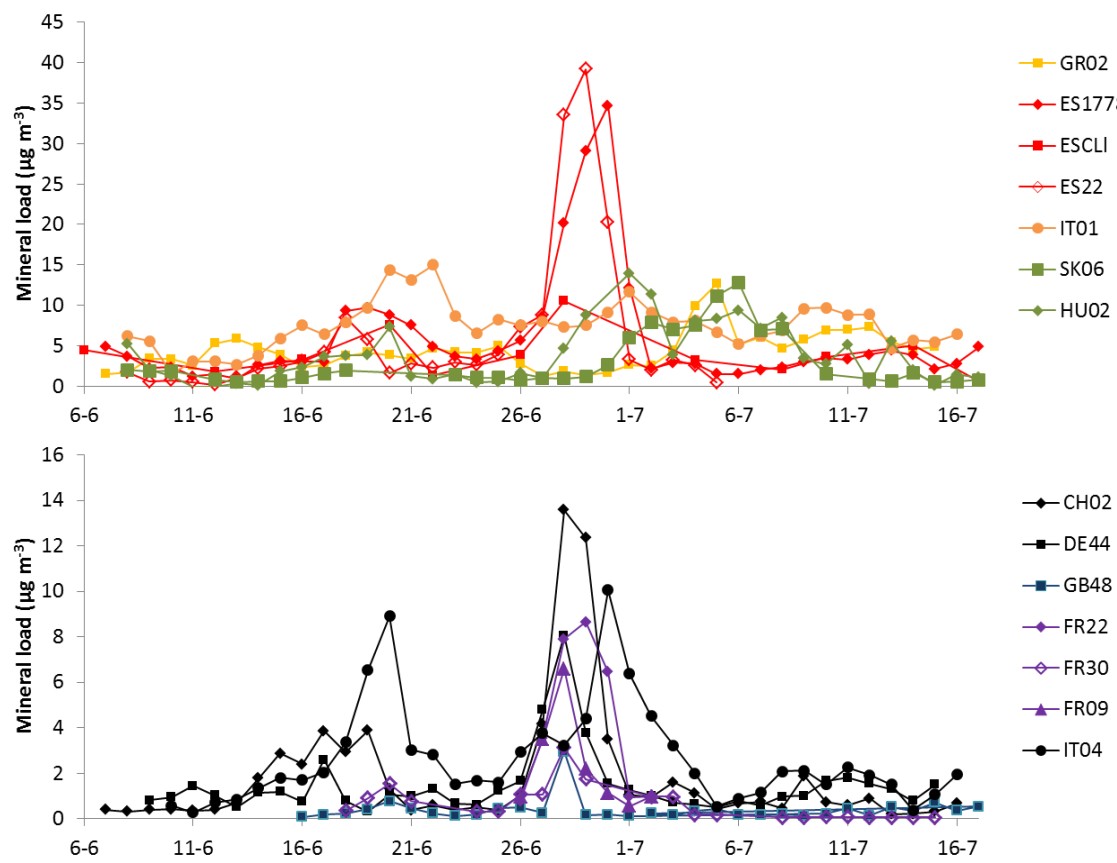

**Figure 8: Time series of mineral dust concentrations during the June/July 2012 EMEP IMP at sites affected by Saharan dust outbreaks. Two major dust episodes were observed in June; in both cases, the dust plume moved along a west to east transect.**

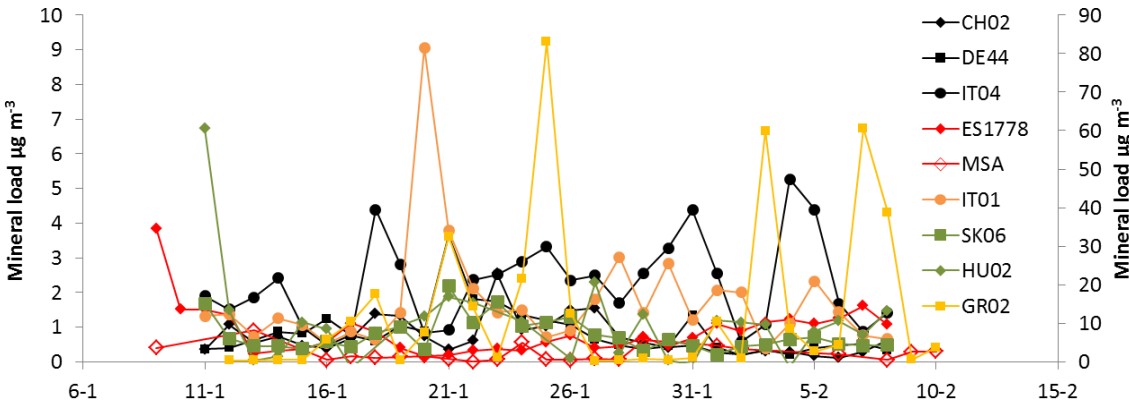

**Figure 9: Time series of mineral dust concentrations during the January/February 2013 EMEP IMP at selected sites affected by the Saharan dust outbreaks. Levels of mineral dust at GR02 (right axis) reflect the impact of short but intense dust pulses.**





**Figure 10: Average composition (absolute values –μg m⁻³, top- and relative contribution -%, bottom) of mineral dust at selected sites when affected by African dust episodes. Results from both summer and winter (GR02-W) IMPs are included.**





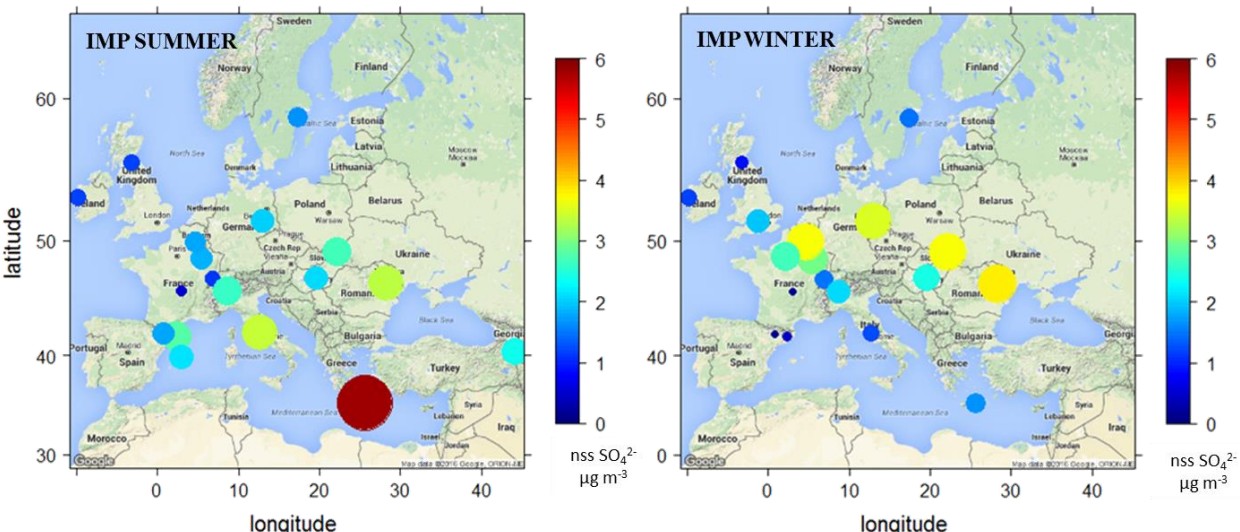

**Figure 11: Spatial distribution of the mean nss-sulphate concentrations (µg m$^{-3}$) during the summer 2012 and winter 2013 EMEP IMPs. The diameter of the circles is proportional to the concentrations.**

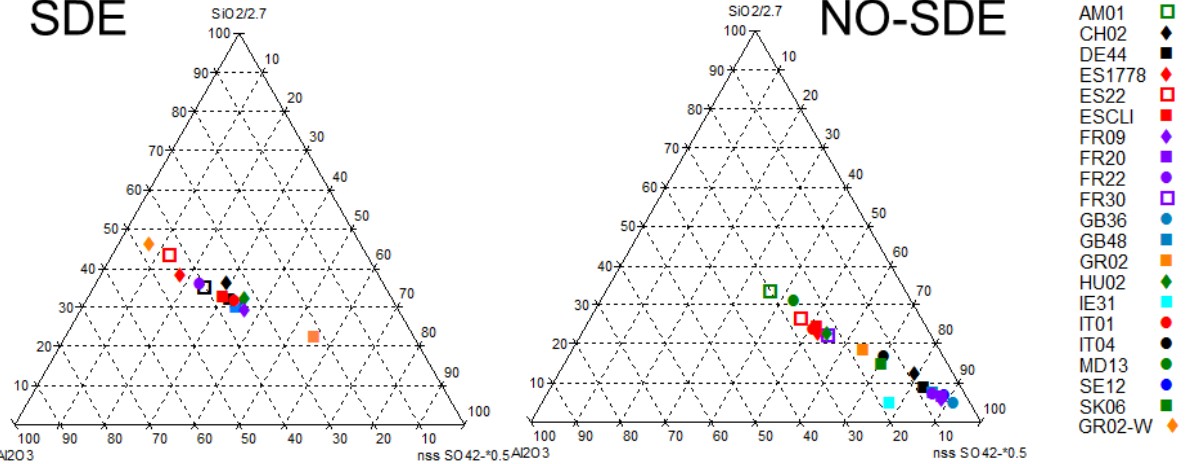

**Figure 12: Ternary diagram for major mineral dust components (SiO$_2$ and Al$_2$O$_3$) and SO$_4^{2-}$ for days with (left, SDE) and without impact of SDE (right, NO-SDE). Orange: GR02; red: SW and SC sites; purple: Central Western Europe; black: CE sites; blue; northern and Atlantic sites; green: eastern sites; empty symbols: high altitude sites.**





**Figure 13: Spatial distributions of the mean concentrations (in ng m⁻³) of Ti, V, and Ni, determined at each site during the summer and the winter EMEP IMPs. The diameter of the circles is proportional to the concentrations.**





**Figure 14: Spatial distributions of the mean concentrations (in ng m$^{-3}$) of Cr, As, and Cu, determined at each site during the summer and the winter EMEP IMPs. The diameter of the circles is proportional to the concentrations.**