# Peer review of "Geochemistry of PM10 over Europe during the EMEP intensive measurement periods in summer 2012 and winter 2013"

_Atmospheric Chemistry and Physics, 2016_

## Referee Comment (RC1) · J. Prospero (Referee) · 18 Mar 2016

General Comments: The authors present the results obtained in the third intensive measurement period (IMP) which took place in the summer of 2012 and winter 2013. PM10 filter samples were concurrently collected at 20 regional background sites across Europe and were subsequently analyzed for their mineral dust content. Because of the uniformity in protocols, they were able to view the data in a coherent way which allowed them to associate composition and concentration patterns to specific generic source types. Mineral dust was a special focus of the study. Dust concentrations were greatest in the southern and easternmost countries, accounting for 20-40% of PM10. This regional impact is largely attributed to Saharan dust outbreaks which were responsible

for the high summer dust loadings at western and central European sites. The spatial distribution of some components and metals reveals the influence of specific anthropogenic sources on a regional scale. Particularly notable was the identification of the impact of shipping emissions (V, Ni, and SO4=) in the regions bordering the Mediterranean. They were also able to identify the impacts of metallurgy sources (on Cr, Ni, and Mn), and coal combustion (As, Se, and SO4=), and traffic (Cu).

The paper highlights the role that mineral dust might play in air quality issues and, thus, human health. On the basis of this paper, it is clear that in some European regions African dust could be an important issue in air quality. However it also shows that mineral dust is a significant contributor in other regions that are not likely to be heavily affected by African dust but rather that local and regional dust sources are significant. Because of the strong focus of this paper on dust and the coherent data set obtained over a large area, it is a unique contribution to the field of air quality.

The paper presents a great deal of information that can make it difficult to see linkages and associations. In general the authors do a good job in identifying such relationships although in some sections this could possibly be done more clearly. They provide many excellent graphical products which facilitate interpretation.

One aspect that is missing from the paper is the sense of the larger context - how the European results compare with similar studies in other regions. There is some review of this aspect in the introductory sections but their is no follow up on this with their own results.

To provide a larger context, perhaps the authors give discuss how the dust activity in the IMP periods compares with dust activity in general during the year and how this year (i.e., experimental "year") compares with other years. This links their work to the general question of variability (and of course to the issue of climate change). I am not suggesting that these issues have to be addressed in any depth, but they might be touched on to provide a sense of scale.

In general the paper is well organized and clearly and succinctly written. It makes a significant contribution to the field of aerosol properties and air quality. The data set are unique in their coverage and coherence.

There are no major problems with this paper and it warrants publication.

Specific Comments: In the section beginning at about 238 and in Fig. 8 and Fig. 9, they discuss the mean levels of dust concentrations measured at various sites. I was surprised that concentrations were not higher and the time-spread of the events was relatively short. They point out for example (ca. line 381) that at the Spanish sites the contribution of Saharan mineral dust accounts for two exceedances of the WHO 24-hour guideline of 50 $\mu$gPM10 m-3. It is interesting to compare these data with the measurements of PM10 in the Caribbean [Prospero et al., Global Biogeochemical Cycles, 2014] where the PM10 exceedances are frequent and driven by African dust. At Caribbean sites the guideline is exceeded on about 10% of the days on an annual basis and as much as 20 - 35% of the days during the peak dust months.

In Section 3.2.2 Mineral dust: 270: "In Eastern Europe, where Saharan dust events did not impact PM levels, a local/regional dust source is deduced." I would have expected some impact. Models and satellite products do show dust events penetrating the region but perhaps the African dust gets lost in the regional soil "noise".

350: 3.3 Mineral dust contribution: impact of Saharan dust events Here the authors summarize and discuss the dust loads measured across the network of stations. As I noted above, the concentrations of dust in the Caribbean are higher than this. I wonder how much of the dust at the European sites is greater than PM10. I note for example that the two big dust events (June-July 2012 and Jan 2012 as shown in Fig. S3) are low-level transport events. I would suspect that there is a considerable component above 10um diameter. Are there any measurements of the complete size distribution either during the experiment periods or obtained at other times? Although this might not be relevant to the discussion of PM10 it might provide some insights on the general

nature of the impact of African dust on health.

The authors do a commendable job in providing a good array of figures (in the paper and in the supplement) that summarize and synthesize the results of their analyses. The ternary diagrams are particularly effective in this regard. They show for example that with respect to $SiO_2$, $Al_2O_3$ and $Fe_2O_3$ (Fig. S5), in dust events the average composition was almost identical for all sites, indicating a similar silico-aluminous composition. The large variability in the contribution of CaO stands out. They relate this to changes in source areas. The uniformity in the composition of long-range-transported African dust has been previously noted, e.g., Trapp et al., Marine Geochem (2010). However I would expect that the European sites might see more variability in trace species in their dust because they are more likely to be impacted by dust from specific sources (as seen in some of the satellite images in Fig. SXXX). In contrast, by the time dust crosses the Atlantic to the Caribbean, the dust from different sources will have been more mixed.

456 et seq.: This section and the discussion about dust-Ca-SO4= is not clear. I would not expect to see much primary gypsum. But I would expect to see varying amounts of gypsum produced from reactions with SO2/SO4=. I see what they are trying to do in this section but it does not come across clearly. What is the conclusion? That there is little or no primary gypsum? Figure S7 is very confusing. How did they decide to draw regressions between specific points as they did. Maybe the explanation is buried in the text. Some text should be provided in the caption.

495: Section 4. Conclusions This section is a straight-forward recapitulation of the major points of the paper. But as I stated in the general comments, I would have liked to have seen more in the way of broadening the perspective of the results, to place the results in the larger picture of dust-health issues. I recognize that the paper is by its nature focused on a specific objective. But it would be nice to round it out with a broader context.

Technical Corrections. Figure S5. What is "CadO" in the top right ternary diagram? Some other aspects of the diagrams also should be explained such as the factors used, e.g., "CadO dust*1.2"; "K2O dust*3.5".

Figure S8 is very complex. But the authors point out the very significant difference in V and Ni over the Mediterranean region as contrasted to other regions. This stands out nicely in the graphs, complex as they are.

Figure 10 caption: "Figure 10: Average composition (absolute values $-\mu$g m-3, top- and relative contribution -%, bottom) of mineral dust at selected sites when affected by African dust episodes. Results from both summer and winter (GR02-W) IMPs are included." Better: Figure 10: Average CONCENTRATION OF MINERAL DUST COMPONENTS (absolute values $-\mu$g m-3, top-) and THE relative contribution of components (%, bottom) IN mineral dust at selected sites when affected by African dust episodes. Results from both summer and winter (GR02-W) IMPs are included.

---

## Referee Comment (RC2) · Anonymous Referee #3 · 23 Mar 2016

This manuscript presents and discusses the composition of PM10 during intensive measurement periods in summer 2012 and winter 2013 for 20 rural background sites across Europe. The emphasis is placed on the mineral dust component, but data for several metals and metalloids are also given. Most of the data were obtained by particle-induced X-ray emission (PIXE). The study clearly shows that PIXE can provide a very valuable contribution in atmospheric aerosol research. The impact on mineral dust, which was often attributed to Saharan dust, on the PM10 aerosol and the spatial variation and seasonality therein are well discussed.

As discussed in detail below, the manuscript is on a few occasions unclear and it has some other weaknesses. There are also some problems with the references. Revision

is needed before this manuscript can be published in ACP.

Specific comments:

1. There are several gratuitous statements within the text with regard to the mineral dust and the non-crustal and non-sea-salt elements, which are not backed up by appropriate references. This is for example the case for the statement in line 66 on the composition of natural mineral dust and in lines 73-74 on the anthropogenic sources of mineral dust, for the attribution of the nssK to aluminosilicates (or to clays, as indicated in line 331) and to biomass combustion in lines 317-318, for the attribution of SO2 to combustion processes in line 456 and for the attribution of part of the nss-sulphate to marine biogenic and volcanic emissions in line 457. Also, which combustion processes do the authors have in mind as source of SO2? Other examples of occasions with lacking references are given below.

2. Lines 48 and 587: Se is mentioned here (respectively in the Introduction and in the Conclusions) and concentration data for this element are given in Tables S3 and S4. However, this element is not discussed at all in the main part of the text. I suggest that it be discussed.

3. Line 61: "Engelstaedter et al., 2006" is not in the Reference list. On the other hand, there is "Engelstaedter et al., 2009" in that list to which no reference is made within the text.

4. Lines 107 and 153: "Lucarelli et al., 2010" is not in the Reference list. On the other hand, there is " Lucarelli et al., 2011" in that list to which no reference is made within the text.

5. Lines 168-169: It is somewhat unclear for which elements the concentrations were close to the analytical detection limit; besides, in which technique were those elements close to the detection limit, in PIXE or in IC-AES/MS or in both techniques?

6. Line 188: A literature reference is needed for the Na/Cl ratio of 0.56 in sea water.

7. Lines 280 and 285: A literature reference is needed for the attribution of (part of the) Al, Si and Fe to a clay-dominated source and to illite.

8. Lines 292-296: Literature references are needed for these statements on Ca.

9. Lines 308-309: A literature reference is needed for this statement on nssMg.

10. Line 381: There is something missing in between "during" and "account".

11. Line 452: What is the basis for attributing part of the K2O to biomass burning? It is known that KCl is a form of K that is present in pyrogenic particles.

12. Line 519: As has besides coal combustion other sources that are of similar or even higher importance, such as Pb production and especially Cu-Ni production (Nriagu and Pacyna, 1988).

13. Line 573: It is unclear what MSY and MSC denote.

14. Figure 12: It is unclear what SW, SC and CE denote.

15. Technical and other minor corrections:

- line 59: the "-1" of "year-1" should be in superscript.

- line 92: the "-2" of "m-2" should be in superscript.

- line 97: subscripts and superscripts are needed for "SO42-, NO3-, NH4+".

- line 141: replace "additionally to" by "addition for".

- line 157: replace "andboth" by "and both".

- line 168: replace "S2in" by "S2 in".

- line 259: replace "10%PM10" by "10% of PM10".

- line 334: replace "SDE in" by "SDEs in".

- line 382: replace "$\mu$gPM10" by "$\mu$g PM10".

- line 430: replace "see Fig.s" by "see Figs.".

- line 440: replace "but whereas" by "whereas".

- line 471: replace "contribution of" by "contributions of".

- line 476: replace "indicate a low" by "indicating a low".

- line 483: replace "Concentrations" by "The concentrations".

- line 496: replace "with Nickel" by "with nickel".

- line 515: replace "see Fig.s" by "see Figs.".

- line 519: replace ".As shown" by ". As shown".

- line 530: replace "see Fig.s" by "see Figs.".

- titles of journal articles in the Reference list should be in lower case instead of in Title Case (e.g., for Aluko and Noll, 2006).

- for authors with two initials in the Reference list, there should be a space between the two initials (e.g., in lines 624. 637, 680).

- line 614: replace "L. C. L. C." by "L. C.".

- line 640: replace "Health 2012;" by "Health,".

- line 648: replace "J Geophys Res 114" by "J. Geophys. Res., 114".

- line 681 replace "U S A 108:" by "U S A, 108, ".

- line 684: replace "andGanor" by "and Ganor".

- line 691: replace "B 318, (2014)" by ", B 318,".

- line 692: replace "B 363" by ", B 363".

- line 695: replace "Environ Health Perspect 2011;" by "Environ. Health Perspect.,".

- line 732: replace "Rev Geophys 40:1002" by "Rev. Geophys., 40, 1002".

- Table 1: it is unclear what the "P" in the Filter column denotes; presumably "Partisol"; in any case, it should be indicated what it denotes.

- Caption of Figure 8: replace "proportional to the concentrations" by "proportional to the concentrations and percentages".

Reference

J. O. Nriagu and J. M. Pacyna, Quantitative assessment of worldwide contamination of air, water and soils by trace metals, Nature, 333, 134-139, 1988.

For the Supplementary Information:

- Heading of Tables S3 and S4: the "-3" of "m-3" should be in superscript.

- Caption of Figure S1: it is unclear what MSY denotes.

- Caption of Figure S4: replace "proportional to the concentrations" by "proportional to the concentrations and percentages".

---

## Author Comment (AC1) · 20 Apr 2016

Geochemistry of PM10 over Europe during the EMEP intensive measurement periods in summer 2012 and winter 2013 – ACP2106-42

**Reply to Prof. Joseph Prospero.**

*Authors thank Professor Joseph Prospero for his valuable comments. We have taken into consideration his suggestions which have enabled us to considerably improve the manuscript. In the paragraphs below we detail the changes addressed.*

**General Comments:**

**The authors present the results obtained in the third intensive measurement period (IMP) which took place in the summer of 2012 and winter 2013. PM10 filter samples were concurrently collected at 20 regional background sites across Europe and were subsequently analyzed for their mineral dust content. Because of the uniformity in protocols, they were able to view the data in a coherent way which allowed them to associate composition and concentration patterns to specific generic source types. Mineral dust was a special focus of the study. Dust concentrations were greatest in the southern and easternmost countries, accounting for 20-40% of PM10. This regional impact is largely attributed to Saharan dust outbreaks which were responsible for the high summer dust loadings at western and central European sites. The spatial distribution of some components and metals reveals the influence of specific anthropogenic sources on a regional scale. Particularly notable was the identification of the impact of shipping emissions (V, Ni, and SO4=) in the regions bordering the Mediterranean. They were also able to identify the impacts of metallurgy sources (on Cr, Ni, and Mn), and coal combustion (As, Se, and SO4=), and traffic (Cu).**

**The paper highlights the role that mineral dust might play in air quality issues and, thus, human health. On the basis of this paper, it is clear that in some European regions African dust could be an important issue in air quality. However it also shows that mineral dust is a significant contributor in other regions that are not likely to be heavily affected by African dust but rather that local and regional dust sources are significant. Because of the strong focus of this paper on dust and the coherent data set obtained over a large area, it is a unique contribution to the field of air quality.**

**The paper presents a great deal of information that can make it difficult to see linkages and associations. In general the authors do a good job in identifying such relationships although in some sections this could possibly be done more clearly. They provide many excellent graphical products which facilitate interpretation.**

**One aspect that is missing from the paper is the sense of the larger context - how the European results compare with similar studies in other regions. There is some review of this aspect in the introductory sections but there is no follow up on this with their own results.**

To provide a larger context, perhaps the authors give discuss how the dust activity in the IMP periods compares with dust activity in general during the year and how this year (i.e., experimental "year") compares with other years. This links their work to the general question of variability (and of course to the issue of climate change). I am not suggesting that these issues have to be addressed in any depth, but they might be touched on to provide a sense of scale. In general the paper is well organized and clearly and succinctly written. It makes a significant contribution to the field of aerosol properties and air quality. The data set are unique in their coverage and coherence.

There are no major problems with this paper and it warrants publication.

Specific Comments:

1. In the section beginning at about 238 and in Fig. 8 and Fig. 9, they discuss the mean levels of dust concentrations measured at various sites. I was surprised that concentrations were not higher and the time-spread of the events was relatively short. They point out for example (ca. line 381) that at the Spanish sites the contribution of Saharan mineral dust accounts for two exceedances of the WHO 24-hour guideline of 50 µgPM10 m-3. It is interesting to compare these data with the measurements of PM10 in the Caribbean [Prospero et al., Global Biogeochemical Cycles, 2014] where the PM10 exceedances are frequent and driven by African dust. At Caribbean sites the guideline is exceeded on about 10% of the days on an annual basis and as much as 20 - 35% of the days during the peak dust months.

*Reply*

*Yes, surprisingly, the number of exceedances can be considered as low if compared with other more distant areas also impacted by Saharan dust outbreaks such as Barbados, where frequent exceedances of the PM10 daily limit value occur related to the Saharan dust outbreaks (Prospero et al 2014). This is commented in the conclusion section in the new version of the manuscript (see below).*

In Section 3.2.2 Mineral dust: 270: "In Eastern Europe, where Saharan dust events did not impact PM levels, a local/regional dust source is deduced." I would have expected some impact. Models and satellite products do show dust events penetrating the region but perhaps the African dust gets lost in the regional soil "noise".

*Reply*

*Yes we have modified this sentence. The impact of the Saharan dust event in Eastern Europe cannot be discarded but, the high contribution of regional/local dust, as deduced from the elevated concentrations and from the geochemical signatures, obscures the evidence of the SDEs.*

2. 350: 3.3 Mineral dust contribution: impact of Saharan dust events Here the authors summarize and discuss the dust loads measured across the network of stations. As I noted above, the concentrations of dust in the Caribbean are higher than this. I wonder how much of the dust at the European sites is greater than PM10. I note for example that the two big

**dust events (June-July 2012 and Jan 2012 as shown in Fig. S3) are low-level transport events. I would suspect that there is a considerable component above 10um diameter. Are there any measurements of the complete size distribution either during the experiment periods or obtained at other times? Although this might not be relevant to the discussion of PM10 it might provide some insights on the general nature of the impact of African dust on health.**

*Reply*

*Unfortunately, there were no measurements of the coarser fraction (>10 µm) during the sampling periods. Actually, in recent years, more studies on the impact of Saharan dust on air quality in Europe have focused on PM10, although, as far as the authors are aware, there are not many studies simultaneously measuring PM10 and TSP during Saharan dust events. A study carried out in the Iberian Peninsula (Rodriguez et al., 2001) concluded that TSP concentrations could be considered as representative of the regional PM10 levels in the rural environment during Saharan dust events, indicating a relatively low contribution of the coarser fraction (>10µm). However, simultaneous measurements of PM10 and TSP were not performed at any site in this study by Rodriguez et al. Moreover, the size ratio may vary depending on the source and transport scenario ( e.g. depending on the altitude of transport) so that a significant contribution of coarse PM cannot be discarded.*

*As regards the health effects of Saharan dust, most studies carried out in Europe have focused on PM10, PM2.5 and/or PM2-5-10, as summarized in the review by Karanasiou et al., 2012. These authors concluded that the health impact of Saharan dust outbreaks needs to be further explored. Therefore, the impact of Saharan dust events on TSP and health outcomes of the coarser fraction (PM >10µm) should be further investigated.*

*References (already in the list)*

*Rodríguez, S., Querol, X., Alastuey, a., Kallos, G. and Kakaliagou, O.: Saharan dust contributions to PM10 and TSP levels in Southern and Eastern Spain, Atmos. Environ., 35(14), 2433–2447, doi:10.1016/S1352-2310(00)00496-9, 2001.*

*Karanasiou, A., Moreno, N., Moreno, T., Viana, M., de Leeuw, F. and Querol, X.: Health effects from Sahara dust episodes in Europe: literature review and research gaps., Environ. Int., 47, 107–14, doi:10.1016/j.envint.2012.06.012, 2012.*

**3. The authors do a commendable job in providing a good array of figures (in the paper and in the supplement) that summarize and synthesize the results of their analyses. The ternary diagrams are particularly effective in this regard. They show for example that with respect to SiO2, Al2O3 and Fe2O3 (Fig. S5), in dust events the average composition was almost identical for all sites, indicating a similar silico-aluminous composition. The large variability in the contribution of CaO stands out. They relate this to changes in source areas. The uniformity in the composition of long-range-transported African dust has been previously noted, e.g., Trapp et al., Marine Geochem (2010). However I would expect that the European sites might see more variability in trace species in their dust because they are more likely to be impacted by dust from specific sources (as seen in some of the satellite images in Fig. SXXX).**

**In contrast, by the time dust crosses the Atlantic to the Caribbean, the dust from different sources will have been more mixed.**

*Reply*

*Yes, the homogeneity observed for dust composition during the summer episodes is surprising. The following sentence, comparing this result with previous results obtained at more distant areas, such as Barbados or Miami, is included in the text, in the conclusion section:*

*"These results are consistent with those obtained fromdust collected on Barbados during Saharan dust events, showing a high uniformity of dust composition that was attributed to the mixing during transport of dust originating from different source areas (Prospero and Lamb 2003, Trapp et al., 2010, Muhs et al., 2014)."*

*As regards variations in CaO contribution: the higher contributions of CaO observed during the winter IMP, as compared with the summer IMP, were attributed to differences in source area. However, during the summer IMP differences in CaO relative contributions at the different sites were attributed to two different causes: 1) higher contribution of local Ca in certain locations (IT01 and ESCLl); 2) preferential deposition of coarse calcium carbonate particles, resulting in higher contributions of CaO in the southern sites. This has been clarified in the revised text.*

*References added:*
*Prospero, J. M. and Lamb, P. J.: African droughts and dust transport to the Caribbean: climate change implications. Science 302:1024–1027, 2003.*
*Trapp, J. M., Millero, F. J. and Prospero, J. M.: Temporal variability of the elemental composition of African dust measured in trade wind aerosols at Barbados and Miami, Mar. Chem., 120(1-4), 71–82, doi:10.1016/j.marchem.2008.10.004, 2010.*
*Muhs, D. R., Prospero, J. M., Baddock, M. C., Gill, T. E., Muhs, D. R., Prospero, J. M., Baddock, M. C. and Gill, T. E.: Identifying Sources of Aeolian Mineral Dust: Present and Past, , in: Knippertz, P. and Stuut, J. W. (Eds): Mineral Dust: A Key Player in the Earth System, Springer Science Business Media Dordrecht 2014.*

**4. 456 et seq.: This section and the discussion about dust-Ca-SO4= is not clear. I would not expect to see much primary gypsum. But I would expect to see varying amounts of gypsum produced from reactions with SO2/SO4=. I see what they are trying to do in this section but it does not come across clearly. What is the conclusion? That there is little or no primary gypsum? Figure S7 is very confusing. How did they decide to draw regressions between specific points as they did. Maybe the explanation is buried in the text. Some text should be provided in the caption.**

*Reply*

Yes, the discussion was not clear enough; Figure S7 has been deleted. The contribution of primary gypsum is very minor, as deduced from low concentrations at mountainous sites. The relative concentrations of sulfate increase with distance indicating that it can attributed to the contribution of secondary anthropogenic sulfate.

**5. 495: Section 4. Conclusions This section is a straight-forward recapitulation of the major points of the paper. But as I stated in the general comments, I would have liked to have seen**

**more in the way of broadening the perspective of the results, to place the results in the larger picture of dust-health issues. I recognize that the paper is by its nature focused on a specific objective. But it would be nice to round it out with a broader context**

*Reply*

*We have tried to improve this section. We have included a sentence about the impact of African dust in Europe, its contribution to PM10 and the inter-annual variability of these contributions. The results are compared to those obtained in Barbados (Prospero et al 2014).*

*We have also included in this section the discussion regarding the homogeneity in composition of dust during the African episodes.*

*We have also highlighted the contribution of regional / local sources of mineral dust to PM10 in southern and eastern areas of Europe, accounting for 10-25% of PM10 during the NON-SDE periods.*

*Reference added:*

*Prospero, J. M., Collard, F.-X., Molinié, J. and Jeannot, A.: Characterizing the annual cycle of African dust transport to the Caribbean Basin and South America and its impact on the environment and air quality, Global Biogeochem. Cycles, 28(7), 757–773, doi:10.1002/2013GB004802, 2014.*

**Technical Corrections.**

**Figure S5. What is "CadO" in the top right ternary diagram? Some other aspects of the diagrams also should be explained such as the factors used, e.g., "CadO dust\*1.2"; "K2O dust\*3.5".**

*Reply*

*This was a mistake. CadO corresponded to CaO. It has been corrected.*

*As regards for the estimation of the factors used in the ternary diagrams: The average concentrations for each element or specie were calculated considering all sites. Then, the average concentration of each element/specie was multiplied by a factor in order to obtain a similar average concentration value for all the elements/species. To this end the average concentration of Al2O3 was taken as a reference. Thus the concentrations adjusted to place the average composition in the center of the triangle for each diagram.*

**Figure S8 is very complex. But the authors point out the very significant difference in V and Ni over the Mediterranean region as contrasted to other regions. This stands out nicely in the graphs, complex as they are.**

*Reply*

*Yes we know that it is complex but we think is the best way to show the differences.*

**Figure 10 caption:** "Figure 10: Average composition (absolute values –µg m-3, top and relative contribution -%, bottom) of mineral dust at selected sites when affected by African dust episodes. Results from both summer and winter (GR02-W) IMPs are included." Better: Figure 10: Average CONCENTRATION OF MINERAL DUST COMPONENTS (absolute values –µg m-3, top-) and THE relative contribution of components (%, bottom) IN mineral dust at selected sites when affected by African dust episodes. Results from both summer and winter (GR02-W) IMPs are included.

*DONE*

---

## Author Comment (AC2) · 20 Apr 2016

**Geochemistry of PM10 over Europe during the EMEP intensive measurement periods in summer 2012 and winter 2013 – ACP2106-42**

**Reply to referee 3.**

*The authors thank the reviewer for the detailed review of the manuscript. The changes suggested have been addressed. We think that the manuscript has greatly improved after these changes.*

**This manuscript presents and discusses the composition of PM10 during intensive measurement periods in summer 2012 and winter 2013 for 20 rural background sites across Europe. The emphasis is placed on the mineral dust component, but data for several metals and metalloids are also given. Most of the data were obtained by particle-induced X-ray emission (PIXE). The study clearly shows that PIXE can provide a very valuable contribution in atmospheric aerosol research. The impact on mineral dust, which was often attributed to Saharan dust, on the PM10 aerosol and the spatial variation and seasonality therein are well discussed.**

**As discussed in detail below, the manuscript is on a few occasions unclear and it has some other weaknesses. There are also some problems with the references. Revision C1 is needed before this manuscript can be published in ACP.**

**Specific comments:**

**1. There are several gratuitous statements within the text with regard to the mineral dust and the non-crustal and non-sea-salt elements, which are not backed up by appropriate references. This is for example the case for the statement in line 66 on the composition of natural mineral dust and in lines 73-74 on the anthropogenic sources of mineral dust, for the attribution of the nssK to aluminosilicates (or to clays, as indicated in line 331) and to biomass combustion in lines 317-318, for the attribution of SO2 to combustion processes in line 456 and for the attribution of part of the nss-sulphate to marine biogenic and volcanic emissions in line 457. Also, which combustion processes do the authors have in mind as source of SO2? Other examples of occasions with lacking references are given below.**

*Reply*

*We agree with the referee. References have been added in order to support the statements:*

*L66*

*Natural mineral dust mainly consists of silicate, carbonate, phosphate and oxide/hydroxide minerals derived from the erosion and weathering of rocks and soils (Moreno et al., 2008, Scheuvens et al., 2013). Mineralogical characterization of desert dust aerosols in Northern Africa carried out during the SAMUM campaign showed that the major constituents of the*

*aerosol were quartz, potassium feldspar, plagioclase, calcite, hematite and the clay minerals illite, kaolinite and chlorite (Kandler et al., 2009, Scheuvens, et al., 2011).*

*L73-74*

*Mineral dust particles are also emitted by anthropogenic sources, such as agricultural activities, construction sites, mining, certain industrial activities such as the cement and ceramic industries, and road dust resuspension (Zender, et al., 2004).*

*L317*

*Non-sea-salt potassium (nssK) has a major aluminium-silicate affinity and may be present in minerals such as K-feldspars, muscovite and illite and clays (Scheuvens et al., 2013), but can also be emitted during biomass combustion (Andreae, 1983, McMeeking et al, 2009) .*

*L331*

*For those sites with a high correlation (R2>0.9) between nssK and Al, we consider that nssK is mostly associated with aluminium silicates. Given the low nssK-Al, the presence of illite or muscovite is likely (Scheuvens et al., 2011).*

*L456 & L457*

*Non-sea-salt sulfate is a major secondary component formed by the oxidation of SO2, mainly emitted by anthropogenic sources such as fossil-fuel combustion processes and metal smelters, as well as natural sources. Moreover, non-sea-salt sulfate (nssSO42-) may have a minor mineral association, mainly as coarse gypsum, and can also be released from marine biogenic and volcanic emissions (Bates el al., 1992 and references therein).*

**2. Lines 48 and 587: Se is mentioned here (respectively in the Introduction and in the Conclusions) and concentration data for this element are given in Tables S3 and S4. However, this element is not discussed at all in the main part of the text. I suggest that it be discussed.**

*Reply*

*The data have been revised and reinterpreted (see comment 12 below). These sentences have been changed by : "….high temperature processes (As, Pb, and $SO_4^{2-}$) in Eastern countries,…".*

*Thanks to referee 3, Tables 3 and 4 were checked. We realized that average concentrations of some elements (Cu ,Zn, As, Se, Br, Rb, Sr, Zr, Mo, Ba, Pb) for sites IE31, FR22, FR09, ES22 were shifted in the Table S3. We corrected the Table.*

**3. Line 61: "Engelstaedter et al., 2006" is not in the Reference list. On the other hand, there is "Engelstaedter et al., 2009" in that list to which no reference is made within the text.**

*Reply*

*- Reference for "Engelstaedter et al., 2006 has been included in the reference list and "Engelstaedter et al., 2009" has been removed*

**4. Lines 107 and 153: "Lucarelli et al., 2010" is not in the Reference list. On the other hand, there is " Lucarelli et al., 2011" in that list to which no reference is made within the text.**

*Reply*

*- The correct reference is Lucarelli et al., 2011; is has been replaced in the text*

**5. Lines 168-169: It is somewhat unclear for which elements the concentrations were close to the analytical detection limit; besides, in which technique were those elements close to the detection limit, in PIXE or in IC-AES/MS or in both techniques?**

*Reply*

*The sentence was certainly unclear; it has been modified as follows:*

*Correlations were lower for V, Cr, Pb and Sr ($R2$ = 0.60-0.69; slopes 0.3 to 0.9), and very low for Ni ($R2$ = 0.19, slope 0.4; see Fig. S1 and Table S2 in the Supplementary Information). For all those elements, concentrations were close to the MDL in PIXE:*

*Caption of Table S1 was also modified: "Table S1: Average minimum detection limit (MDL) for the elements measured by PIXE."*

**6. Line 188: A literature reference is needed for the Na/Cl ratio of 0.56 in sea water.**

*Reply*

*Reference "Drever 1997" has been added*

**7. Lines 280 and 285: A literature reference is needed for the attribution of (part of the) Al, Si and Fe to a clay-dominated source and to illite.**

*Reply*

*The reference Scheuvens et al., (2011) has been added*

**8. Lines 292-296: Literature references are needed for these statements on Ca.**

*Reply*

*The text has been modified as follows:*

*$CaSO_4•2H_2O$, and/or anhydrite - $CaSO_4$) or as silico-aluminates (Ca-plagioclase) (Scheuvens et al., 2013). This element is usually related to natural sources (soil resuspension), although it can be emitted by a number of anthropogenic sources, such as road dust and construction activities (Amato et al., 2009). Calcium carbonate may interact with acidic compounds in the atmosphere forming coarse secondary calcium nitrates ($Ca(NO_3)_2$) and calcium sulphates ($CaSO_4•xH_2O$) (Dentener et al., 1996, Krueger et al., 2004, Alastuey et al., 2005, Hwang and Ro, 2006).*

**9. Lines 308-309: A literature reference is needed for this statement on nssMg.**

*Reply*

*The text has been modified as follows:*

*The nssMg may be associated with clays, carbonates (dolomite), aluminium-silicates or salts (Scheuvens et al., 2011).*

**10. Line 381: There is something missing in between "during" and "account".**

*Reply*

*"…dust during the sampling period account…"*

**11. Line 452: What is the basis for attributing part of the K2O to biomass burning? It is known that KCl is a form of K that is present in pyrogenic particles.**

*Reply*

*K2O has been replaced by K*

**12. Line 519: As has besides coal combustion other sources that are of similar or even higher importance, such as Pb production and especially Cu-Ni production (Nriagu and Pacyna, 1988).**

*Reply*

*Yes; this comment is very pertinent. Actually we have revised the data interpretation and identified higher correlations of As with other elements such as Pb. Initially we did not consider the French sites for this analysis and the correlation As/Se was more significant. However, when these sites were considered this correlation decreased. The following sentence has been included in the text:*

*"A similar trend was also observed for pollutants such as Pb (see Tables S3 and S4). Sources of As, Pb, and $SO_2$ (and other volatile pollutants such as Cd, Se, and Hg) are related to high-temperature processes such as coal combustion, roasting and smelting of ores in non-ferrous metal smelters (Pb and Cu-Ni production) and melting operations in ferrous foundries, among others (Pacyna, 1986, and Niragu and Pacyna, 1988)."*

*Corresponding sentences in the abstract and in the conclusion sections were modified (as indicated in comment 2.*

**13. Line 573: It is unclear what MSY and MSC denote.**

*Reply*

*MSY and MSC refer to the Montseny and Montsec sites. They have been replaced by ES1778 and ES22, which are the EMEP codes used in the manuscript.*

**14. Figure 12: It is unclear what SW, SC and CE denote.**

*Reply*

*Caption has been modified, replacing "SW and SC" by "Southwestern and Central Southern Europe" and CE by "Central Europe"*

*Figure 12: Ternary diagram for major mineral dust components (SiO2 and Al2O3) and SO42- for days with (left, SDE) and without impact of SDE (right, non-SDE). Orange: GR02; red: sites; purple: Central Western Europe; black: Central Europe sites; blue; Northern Europe and Atlantic sites; green: Eastern Europe sites; empty symbols: high altitude sites.*

*This change has also been made for Figures S5, S6, and S10*

**15. Technical and other minor corrections:**

*Reply*

*All the technical corrections have been applied*

**- line 59: the "-1" of "year-1" should be in superscript.** *- DONE*

**- line 92: the "-2" of "m-2" should be in superscript.** *- DONE*

**- line 97: subscripts and superscripts are needed for "SO42-, NO3-, NH4+".** *- DONE*

**- line 141: replace "additionally to" by "addition for".** *- DONE*

**- line 157: replace "andboth" by "and both".** *- DONE*

**- line 168: replace "S2in" by "S2 in".** *- DONE*

**- line 259: replace "10%PM10" by "10% of PM10".** *- DONE*

**- line 334: replace "SDE in" by "SDEs in".** *- DONE*

**- line 382: replace "µgPM10" by "µg PM10".** *- DONE*

**- line 430: replace "see Fig.s" by "see Figs.".** *- DONE*

**- line 440: replace "but whereas" by "whereas".** *- DONE*

**- line 471: replace "contribution of" by "contributions of".** *- DONE*

**- line 476: replace "indicate a low" by "indicating a low".** *- DONE*

**- line 483: replace "Concentrations" by "The concentrations".** *- DONE*

**- line 496: replace "with Nickel" by "with nickel".** *- DONE*

**- line 515: replace "see Fig.s" by "see Figs.".** *- DONE*

**- line 519: replace ".As shown" by ". As shown".** *- DONE*

**- line 530: replace "see Fig.s" by "see Figs.".** *- DONE*

**- titles of journal articles in the Reference list should be in lower case instead of in Title Case (e.g., for Aluko and Noll, 2006).** *- DONE*

**- for authors with two initials in the Reference list, there should be a space between the two initials (e.g., in lines 624. 637, 680).** *- DONE*

**- line 614: replace "L. C. L. C." by "L. C.".** *- DONE*

**- line 640: replace "Health 2012;" by "Health,".** *- DONE*

**- line 648: replace "J Geophys Res 114" by "J. Geophys. Res., 114".** *This references was removed*

**- line 681 replace "U S A 108:" by "U S A, 108, ".** *- DONE*

**- line 684: replace "andGanor" by "and Ganor".** *- DONE*

**- line 691: replace "B 318, (2014)" by ", B 318,".** *- DONE*

**- line 692: replace "B 363" by ", B 363".** *- DONE*

**- line 695: replace "Environ Health Perspect 2011;" by "Environ. Health Perspect.,".** *- DONE*

**- line 732: replace "Rev Geophys 40:1002" by "Rev. Geophys., 40, 1002--** *DONE*

**- Table 1: it is unclear what the "P" in the Filter column denotes; presumably "Partisol"; in any case, it should be indicated what it denotes.** *–"P" refers to Pallflex. Indicated in the caption.*

**- Caption of Figure 8: replace "proportional to the concentrations" by "proportional to the concentrations and percentages".** *– DONE. Change also done in caption of Figures 3, 6 and S4.*

**Reference J. O. Nriagu and J. M. Pacyna, Quantitative assessment of worldwide contamination of air, water and soils by trace metals, Nature, 333, 134-139, 1988.** *– Reference added*

**For the Supplementary Information:**

**- Heading of Tables S3 and S4: the "-3" of "m-3" should be in superscript.** *- DONE*

**- Caption of Figure S1: it is unclear what MSY denotes.** *- DONE*

**- Caption of Figure S4: replace "proportional to the concentrations" by "proportional to the concentrations and percentages".** *- DONE*